# Sparse and Faithful Local Explanations with Piecewise Linear Surrogates

**Yixin Wang** [1]   **Yucheng Dong** [1]

## Abstract

Local post-hoc explanations are widely used to understand black-box models on tabular data, with Local Interpretable Model-agnostic Explanations (LIME) being a popular approach. LIME approximates a black-box model using a sparse linear surrogate in a local neighborhood, implicitly assuming feature-wise linear homogeneity. However, this assumption often fails when local feature effects exhibit heterogeneous or nonlinear behaviors, resulting in unfaithful and unstable explanations. Moreover, LIME relies on a decoupled feature selection procedure that is not aligned with the surrogate modeling objective, further exacerbating instability under local sampling. To address these limitations, we propose PL-LIME, a two-stage sparse local explanation framework that ensures objective consistency across stages. PL-LIME models feature-wise local effects using instance-anchored piecewise linear functions, providing a minimal yet principled extension beyond linear surrogates under a fixed explanation budget. Sparsity is enforced through a decoupled nonnegative shrinkage procedure that directly scales the estimated local effects, improving stability while preserving interpretability. Experiments on synthetic and real-world datasets demonstrate that PL-LIME achieves higher local fidelity and stability, and provides more reliable local explanations that capture finer-grained local effect structures than LIME.

## 1. Introduction

In high-stakes decision-making domains such as finance, healthcare, biology, supply chain management, and public health, machine learning–based predictive models have been widely deployed. To improve predictive performance, many recent approaches employ highly expressive models, including deep neural networks and ensemble methods. However, the decision logic of such models is typically opaque, making their predictions difficult to interpret or trust. This lack of transparency poses significant challenges for regulatory compliance, accountability, and decision reliability in real-world settings (Guidotti et al., 2018; Freitas, 2014) . As a result, developing faithful and human-interpretable explanations for black-box models, particularly post-hoc local explanations at the level of individual predictions (Arrieta et al., 2020), has emerged as a key problem in modern machine learning.

Among model-agnostic local explanation methods, LIME (Local Interpretable Model-agnostic Explanations) (Ribeiro et al., 2016) has emerged as one of the most widely used approaches due to its conceptually simple design and broad applicability across different models. The key idea of LIME is to generate perturbed samples in the neighborhood of an explained instance and fit a simple linear surrogate model to approximate the black-box model's local behavior. The coefficients of the surrogate model are then interpreted as feature importance scores. However, prior studies have shown that LIME often suffers from poor stability and limited local fidelity (Zhang et al., 2019; Zhou et al., 2021; Guidotti et al., 2019; Rahnama & Boström, 2019; Garreau & Luxburg, 2020). Existing work typically attributes these issues to factors such as inappropriate perturbation distributions, weighting schemes, or limitations of the surrogate modeling strategy. From a modeling perspective, LIME relies on an overly coarse definition of locality, implicitly assuming that the local behavior of complex black-box models is homogeneous and can be captured by a single sparse linear function. In practice, local behavior is often heterogeneous even within a small neighborhood, making a single sparse linear approximation biased. Moreover, LIME's sparsity mechanism is enforced through a decoupled linear feature selection procedure whose objective is misaligned with local effect estimation.

To address these issues, numerous extensions of LIME have been proposed, including improved perturbation strategies, refined weighting schemes, and the incorporation of prior knowledge (Tan et al., 2023; Zafar & Khan, 2021; Shankaranarayana & Runje, 2019; Laugel et al., 2018; Zhao et al., 2021; Li et al., 2023; Slack et al., 2021), as well as replac-

---

[1]Business School, Sichuan University, Chengdu, Sichuan, China. Correspondence to: Yucheng Dong <ycdong@scu.edu.cn>.

*Proceedings of the $43^{rd}$ International Conference on Machine Learning*, Seoul, South Korea. PMLR 306, 2026. Copyright 2026 by the author(s).

ing linear surrogates with nonlinear models (Guidotti et al., 2019; van der Waa et al., 2018; Bramhall et al., 2020; Hung & Lee, 2024). While these approaches partially improve stability or local fidelity, most of them still treat the local neighborhood as a single indivisible explanatory unit, failing to explicitly model local structure within the neighborhood. Moreover, prior sparse surrogate methods often suffer from misaligned feature selection (Plumb et al., 2018) and fitting objectives or rely on opaque structural heuristics (Hung & Lee, 2024).

Our key insight is that improving local explanation quality does not hinge on fitting a single, more expressive surrogate model, but rather on refining the definition of locality itself. This motivates modeling the local neighborhood as a collection of finer-grained sub-local regions in which linearity becomes a valid approximation. Crucially, such a refined notion of locality also calls for a sparsification mechanism that is objective-consistent, controllable, and stable, so that feature selection aligns with local effect estimation rather than distorting it.

Building on this insight, we propose PL-LIME, an instance-anchored piecewise linear local explanation framework that models feature-wise local effects using piecewise linear functions under a fixed explanation budget. By partitioning the local neighborhood into multiple input intervals and enforcing linearity within each interval, PL-LIME provides a minimal yet principled extension beyond linear surrogates, explicitly capturing how feature effects vary across sub-local regions through parameters with clear directional and quantitative interpretations. To ensure sparsity remains aligned with this refined locality structure, we introduce a decoupled and objective-consistent two-stage sparsification strategy. Local feature effects are first estimated without sparsity constraints, and sparsity is subsequently enforced via nonnegative shrinkage that operates directly on the estimated effects. We further provide two explanation dimensions. First, we introduce a sparse variance-based feature importance measure to quantify the relative strength of feature contributions. Second, the proposed piecewise linear surrogate yields local feature effect shape functions that characterize how each feature influences the prediction across sub-local regions.

The main contributions of this work are summarized as follows. (i) We identify two coupled structural limitations of LIME, namely overly coarse locality modeling and a sparsification mechanism misaligned with local effect estimation, and show how they jointly lead to unfaithful and unstable local explanations. (ii) We propose PL-LIME, a local explanation framework that refines local explanations through instance-anchored piecewise linear modeling, while preserving the interpretability of linear explanations under a fixed explanation budget. (iii) We introduce an objective-consistent, decoupled sparsification framework that is gener-

ally applicable to additive local surrogate models, where the first-stage additive surrogate can be replaced by alternative additive formulations.

## 2. Background and Limitations of LIME

**Local Linear Explanations with LIME.** Given an instance $x_i$, LIME generates perturbed samples around $x_i$ and assigns proximity-based weights to emphasize locality. A sparse subset of important features is first selected using an $\ell_1$-based feature selection method (e.g., Lasso or LARS), after which a weighted ridge regression is fitted on the selected features to estimate the local surrogate. The resulting linear coefficients are interpreted as local feature importance scores. A key implicit assumption underlying LIME is that the black-box response is locally linear around $x_i$, such that each feature effect can be captured by a single coefficient within the local neighborhood.

**Limitations of the Local Linearity Assumption.** From a feature-wise perspective, this assumption can be restrictive in practice. Even within a small neighborhood, the effect of an individual feature may vary across different subregions when other features are held fixed. As a result, a single linear coefficient may fail to faithfully capture heterogeneous local feature effects. Empirically, we observe that such variations occur frequently across datasets and black-box models, indicating that the local linearity assumption is often violated in practice. This limitation motivates the need for surrogate models that can account for varying local behaviors without enforcing a single linear explanation per feature . The formal definition, measurement of local linearity violations, implementation details, and additional results are provided in Appendix B.1.

## 3. Related Work

Local post-hoc explanation methods for black-box models on tabular data can be broadly categorized by their explanation form, including feature-attribution-based methods and rule-based methods. Feature-attribution methods such as LIME (Ribeiro et al., 2016) and SHAP (Lundberg & Lee, 2017) generate local explanations by assigning a scalar importance score to each feature around the explained instance. LIME fits a linear surrogate on locally perturbed samples, while SHAP computes Shapley values based on average marginal contributions. Although intuitive, these methods summarize local behavior using a single coefficient per feature and do not explicitly characterize how feature effects vary within the local neighborhood.

Several extensions of LIME aim to improve local fidelity and stability by modifying perturbation strategies or weighting schemes (Tan et al., 2023; Zafar & Khan, 2021; Shankaranarayana & Runje, 2019; Laugel et al.,

2018). Other works incorporate global priors, sparsity constraints, or Bayesian inference to enhance explanation consistency (Zhao et al., 2021; Li et al., 2023; Slack et al., 2021). Despite these advances, most approaches retain the core assumption that a single linear surrogate adequately captures local black-box behavior. Some methods adopt more expressive nonlinear surrogates, such as quadratic models (Bramhall et al., 2020), Multivariate Adaptive Regression Splines (MARS) (Hung & Lee, 2024), a mixture regression model enhanced by fused lasso (Guo et al., 2018) and decision trees (Botari et al., 2020; Guidotti et al., 2019) While these methods improve local fidelity, they typically treat the entire neighborhood as a single explanatory unit, without further partitioning or refining the local structure, and thus fail to provide interpretable explanations over distinct subregions of the local neighborhood.

Prior work on feature sparsity for surrogate models can be broadly categorized into linear and nonlinear approaches. LIME performs feature selection by first identifying important features via LASSO or the LARS solution path, followed by linear fitting on the selected subset. For local linear surrogate models, GLIME (Li et al., 2023) introduces an $\ell_1$ sparsity-inducing penalty into the linear regression objective and leverages the LARS solution path to perform feature selection, thereby jointly optimizing local feature effect and sparsity. MAPLE (Plumb et al., 2018) adopts a two-stage strategy in which feature importance scores are computed based on impurity reductions at the root nodes of trees in a random forest, followed by linear fitting on the selected feature subset. For nonlinear surrogate models, feature sparsity is achieved through model-structure constraints or regularization mechanisms, such as decision tree pruning, backward pruning in MARS (Hung & Lee, 2024), or penalties like fused lasso (Guo et al., 2018). Overall, existing sparsity methods either suffer from a mismatch between the objectives of feature selection and model fitting (Ribeiro et al., 2016; Plumb et al., 2018), or tightly couple sparsity with the fitting objective (Li et al., 2023; Hung & Lee, 2024; Guo et al., 2018), while some approaches remain opaque in terms of their feature selection mechanisms, such as decision tree pruning and backward pruning in MARS, limiting interpretability.

In contrast to existing approaches, our work aims to preserve parameter interpretability while making local surrogate models structurally more local. By adopting piecewise linear surrogates, we capture finer-grained local structure in feature effects across different value intervals. Moreover, we propose a sparse surrogate approach that decouples local feature effect estimation from sparsity induction, while ensuring objective alignment across the two stages.

## 4. Piecewise Local Linear Explanations

Building upon LIME and motivated by the presence of heterogeneous local behaviors, we propose a more flexible yet interpretable framework for local surrogate models. Specifically, we consider an additive explanation model of the form

$$f(x) = \sum_{i=1}^{p} \theta_i \, u_i(x_i) + b, \qquad \theta_i \geq 0, \qquad (1)$$

where $u_i(x_i)$ denotes the marginal contribution shape function of feature $i$, and $\theta_i$ is a shrinkage factor that controls whether and to what extent the feature contributes to the explanation.

### 4.1. Modeling Value-Dependent Feature Contributions

The marginal contribution shape function $u_i(x_i)$ is designed to provide a value-dependent explanation of feature $i$ within a local neighborhood, revealing how the feature's contribution evolves as its value deviates from the explained instance and enabling users to identify variation in contributions across different value ranges. To model value-dependent variation in feature contributions transparently, for each continuous feature $x_i$, we construct a piecewise linear marginal shape function $u_i(x_i)$

**Piecewise Linear Function Construction.** The value range of $x_i$ is divided into $m$ consecutive intervals, with knots $k_{i0}, k_{i1}, \ldots, k_{im}$, where $k_{i0}$ and $k_{im}$ correspond to the minimum and maximum values of $x_i$, respectively, and the length of each interval is $r_{i,j} = k_{i,j} - k_{i,j-1}$ for $j = 1, \ldots, m$. For each interval, we define the first-order differences $\Delta f_{i,1}, \ldots, \Delta f_{i,m}$, which represent the change in feature contribution across the interval. The cumulative contribution at knot $k_{ij}$ is $f_{i,k_j} = \sum_{\ell=1}^{j} \Delta f_{i,\ell}$ with $f_{i,k_0} = 0$. For any feature value $x_i$,

$$u_i(x_i) = \sum_{j=1}^{m} \left[ f_{i,k_{j-1}} + \frac{(x_i - k_{i,j-1}) \, \Delta f_{i,j}}{r_{i,j}} \right] I_{i,j}^k(x_i), \tag{2}$$

where $I_{i,j}^k(x_i) = \mathbf{1}_{[k_{i,j-1}, k_{i,j})}(x_i)$. This piecewise linear construction is functionally equivalent to a degree-1 truncated power basis spline, but adopts an interval-wise parameterization in which each parameter corresponds to the total contribution change within a specific interval, thereby enhancing interpretability. (details are provided in the Appendix A.1).

**Interpretability of First-Order Differences.** Each first-order difference $\Delta f_{i,j}$ captures the direction and magnitude of the change in feature contribution over the $j$-th interval (positive for increasing contribution and negative for decreasing). From an interpretability perspective, $\Delta f_{i,j}$ represents the total contribution change within the interval, while the corresponding interval-wise slope is given

by $s_{i,j} = \Delta f_{i,j}/r_{i,j}$. The sign of $\Delta f_{i,j}$ indicates whether the feature increases or decreases the prediction, whereas its magnitude reflects how strongly the feature contribution varies across value intervals. Moreover, such a parameterization naturally enables the incorporation of prior structural constraints, e.g., monotonicity or diminishing marginal effects.

We analyze the behavior of LIME when the true feature-wise local effect is piecewise linear. This analysis explains why representing a local effect with a single coefficient can be insufficient.

When the effect is locally homogeneous, the coefficient fitted by LIME can be interpreted as a local slope. When the effect varies across sub-local intervals, a single slope can only summarize the neighborhood rather than represent each interval-specific effect. The fitted coefficient therefore becomes a kernel-weighted average of interval-wise slopes, which can mask value-dependent local effects. The following proposition formalizes this behavior.

**Proposition 4.1** (LIME under piecewise-linear local effects). *Assume the true marginal function of feature $r$ is piecewise linear with interval-wise slopes $\{s_{r,j}\}_{j=1}^m$ in a neighborhood of the instance being explained. Under Gaussian perturbations and kernel weighting, the population LIME coefficient satisfies*

$$\beta_r^* = \sum_{j=1}^m \omega_{r,j}\, s_{r,j}, \qquad (3)$$

*where $\{\omega_{r,j}\}$ are nonnegative weights that sum to one and depend on the perturbation distribution and the kernel. In particular, if all $s_{r,j}$ are equal (locally linear case), LIME exactly recovers the true slope.*

Proposition 4.1 shows that when the true marginal effect is heterogeneous, LIME collapses multiple interval-wise slopes into a single kernel-weighted average. The detailed statement and proof are deferred to Appendix A.2.

### 4.2. Sparse Feature Selection for Local Explanations

LIME models the local behavior of a black-box predictor around an explained instance $x_0$ using a sparse additive surrogate of the form $\sum_{i=1}^p \theta_i u_i(x_i) + b$, where $\theta_i \in \{0,1\}$ indicates whether the $i$-th feature is selected, In LIME formulation, $u_i(x_i)$ is typically modeled as a linear function in the neighborhood of $x_0$. When feature effects exhibit non-linear behavior, feature selection based on an $\ell_1$-regularized linear model may lead to biased or unstable selections under different local samplings.

Rather than jointly optimizing feature selection and local feature effect estimation, LIME adopts a decoupled two-stage procedure: a sparse feature subset is first identified

using an $\ell_1$-regularized regression such as Lasso or LARS , followed by refitting a ridge regression model on the selected features. However, the objectives optimized in the two stages are not aligned. Despite this limitation, the decoupling strategy plays an important role in local explanation models. By separating feature selection from local feature effect fitting, this design reduces the influence of sparsity constraints on the estimation of local feature effects, compared to approaches that directly impose sparsity within the fitting objective.

**Design Desiderata for Sparse Local Explanations.** These insights motivate two desiderata for sparse local explanation models.

**Desideratum D1: Decoupled Sparsification.** Local effect estimation and sparse feature selection should be performed in two sequential stages.

This separation ensures that sparsity constraints do not distort the estimates of $u_i^*(\cdot)$ obtained during the estimation stage, allowing the model to better capture genuine data patterns. The feature selection stage directly controls the simplicity of the explanation.

**Desideratum D2: Objective Alignment.** Sparse local explanations should enforce sparsity through the same objective that defines the explanation itself. Specifically, sparsification should be achieved by directly constraining or shrinking the fitted local effects, rather than optimizing feature selection and effect estimation under different criteria.

Under the above desiderata, LIME satisfies Desideratum D1, but does not provide guarantees for Desideratum D2. This motivates the design of a new two-stage framework that preserves the advantages of decoupled sparsification while ensuring objective alignment across stages. Specifically, we propose a decoupled yet **objective-consistent two-stage framework**.

In the first stage, we center the response within the local neighborhood of the instance being explained, so that no intercept term is required. We fix $\theta_i = 1$ and focus on estimating value-dependent local feature effects parameterized by the first-order differences, while retaining all features without applying sparsification.

Specifically, we solve the weighted least-squares problem

$$\min_{\{\Delta f_{i,j}\}} \left\| W^{1/2} \left( \sum_{i=1}^p u_i(x_i) - y \right) \right\|_2^2, \qquad (4)$$

where $u_i(x_i)$ is the piecewise linear function defined in Eq. (2). This stage estimates local feature contributions prior to any sparsification. The diagonal weight matrix $W$ encodes sample locality, assigning larger weights to observations closer to the instance being explained.

Letting $\boldsymbol{\Delta f}$ collect all first-order differences $\{\Delta f_{i,j}\}$, the model can be written in matrix form as $\hat{\mathbf{y}} = \mathbf{A}\boldsymbol{\Delta f}$, where $\mathbf{A}$ is the design matrix. Since the objective is a weighted least-squares problem, the solution admits the closed form $\widehat{\boldsymbol{\Delta f}} = (\mathbf{A}^{\top} W \mathbf{A})^{-1} \mathbf{A}^{\top} W \mathbf{y}$. The explicit construction of $\mathbf{A}$ is provided in Appendix B.2.

In the second stage, let $U^*$ denotes the design matrix constructed from the first-stage estimates $\{u_i^*(x_i)\}_{i=1}^p$, and estimate $\theta_i$ by solving

$$\min_{\theta \in [0,\infty)^p} \left\| W^{1/2} \left( U^* \theta - y \right) \right\|_2^2 + \lambda\, c^{\top} \theta, \qquad (5)$$

where $\theta_i$ acts as a non-negative scaling coefficient, $p$ is the number of features, and $c_i$ specifies the penalty weight for feature $i$, set to the number of parameters used to represent its local effect (i.e., $c_i = m_i$ for continuous features and $c_i = 1$ for categorical features). The final prediction of the local interpretable surrogate model is given by $\sum_{i=1}^p \theta_i^*\, u_i^*(x_i),$.

The proposed sparse local explanation method is designed to satisfy the above design principles by introducing nonnegative shrinkage factors $\theta_i$ that are applied multiplicatively to the initial local effect estimates. These factors induce sparsity by shrinking redundant components to zero while preserving components that are close to the true local effects, and thereby ensure that feature selection and effect estimation are derived from the same local surrogate objective rather than from separate optimization criteria.

Beyond this structural property, theoretical guarantees can be established under additional assumptions on the first-stage estimator. In particular, if the initial local effect estimates are sufficiently close to the true local effects, formalized by an $\ell_2$-consistency condition, The proposed method achieves both estimation consistency and variable selection consistency, removing features with zero true effect while recovering the magnitudes of nonzero local effects. This result is stated as a conditional guarantee in the Appendix B.3 and is not required for the validity of the proposed method.

In this work, we instantiate the proposed two-stage framework using a minimal nonlinear local surrogate, termed **PL-LIME (instance-anchored)**, designed to operate under a fixed explanation budget. Our objective is not to approximate the true infinitesimal local behavior of the black-box model, but to expose structured variation in feature contributions at the explanation-relevant scale. Motivated by this goal, for each continuous feature, the local effect function $u_i(x_i)$ is modeled as a piecewise linear function with a single knot anchored at the explained instance $x_{i0}$, yielding two linear segments on either side of $x_{i0}$. Relative to a linear surrogate, this introduces one additional degree of freedom per feature—the minimal increase in flexibility sufficient to disentangle value-dependent local contribu-

tions that are otherwise averaged by LIME. As shown in Proposition 4.1, when marginal effects are heterogeneous, LIME collapses value-dependent feature contributions into a single kernel-weighted slope. PL-LIME follows the same two-stage procedure described above: local effect functions are first estimated without sparsity constraints, followed by second-stage nonnegative shrinkage on the scaling coefficients $\theta$. Unless otherwise specified, PL-LIME refers to this instance-anchored configuration with a single knot per continuous feature.

### 4.3. Sparse Variance-Based Local Feature Importance

We develop a unified variance-based framework for analyzing and quantifying local feature importance in sparse additive surrogate models. Within this framework, we introduce a variance-based feature importance measure for sparse additive local surrogate models.

**Definition 4.2** (Sparse Variance-based Explanation Importance). Consider the fitted sparse additive local surrogate model $\hat{f}(x) = \sum_{i=1}^p \hat{\theta}_i\, \hat{u}_i(x_i)$, where $\hat{u}_i(\cdot)$ denotes the estimated local effect function obtained from the first-stage estimation, and $\hat{\theta}_i$ is the corresponding shrinkage coefficient. the sparse variance-based explanation importance of feature $X_i$ for the instance to be explained $x_0$ is defined as

$$\hat{I}_i(x_0) := \frac{\hat{\theta}_i\, \sigma(\hat{u}_i(X_i))}{\sum_{j=1}^p \hat{\theta}_j\, \sigma(\hat{u}_j(X_j))}, \qquad (6)$$

where $\sigma_{x_0}(\hat{u}_i(X_i))$ denotes the standard deviation, i.e., the square root of the local variance, of $\hat{u}_i(X_i)$ computed under the empirical distribution of the locally generated samples of $X_i$ in the neighborhood of $x_0$.

This measure admits a direct interpretation: for the fitted surrogate model $\hat{f}$, a one-standard-deviation increase in $\hat{u}_i(x_i)$ leads to an approximate change of $\hat{\theta}_i \sigma(\hat{u}_i(X_i))$ in the response.

*Remark* 4.3 (Reduction to LIME under Linear Local Surrogates). When no feature selection is applied, i.e., $\hat{\theta}_i = 1$ for all $i$, and the surrogate function space is linear, under the assumption that the locally sampled features are standardized, i.e., $\sigma(X_i) = c$ for all $i$. Then $\hat{I}_i(x_0) \propto |\beta_i|$, where $\beta_i$ denotes the fitted linear coefficient in LIME.

**Recovering True Local Effect Strength.** We study to what extent different surrogate function spaces can recover the true local effect strength of each feature.

When no feature selection is applied, i.e., $\hat{\theta}_i = 1$ for all $i$, we consider a locally additive representation of a black-box model around the explained instance $x_0$, given by $f(x) = \sum_{i=1}^p u_i^0(x_i) + \varepsilon$, where $u_i^0(\cdot)$ denotes the true local effect of feature $X_i$. We assume each $u_i^0$ is centered under the local perturbation distribution $P_{x_0}$, i.e., $\mathbb{E}_{X_i \sim P_{x_0}}\left[ u_i^0(X_i) \right] = 0$.

**Definition 4.4** (True Effect Strength). The true effect strength of feature $X_i$ is defined as $S_i^0 := \mathrm{Var}\big(u_i^0(X_i)\big) = \|u_i^0\|_{L^2}^2$.

This quantity captures the intrinsic contribution of feature $X_i$ to the model output under local perturbations. Let $H \subset L^2(X_i)$ be a surrogate function space. The $L^2$-projection of $u_i^0$ onto $H$ is defined as $\Pi_H u_i^0 := \arg\min_{h \in H} \mathbb{E}\big[(u_i^0(X_i) - h(X_i))^2\big]$. By the orthogonality property of $L^2$, we have $\|u_i^0\|_{L^2}^2 = \|\Pi_H u_i^0\|_{L^2}^2 + \|u_i^0 - \Pi_H u_i^0\|_{L^2}^2$.

**Definition 4.5** (Surrogate Effect Strength). The surrogate effect strength of feature $i$ under space $H$ is defined as $S_i^H := \|\Pi_H u_i^0\|_{L^2}^2$.

This definition quantifies how much of a feature's true local effect can be recovered by a surrogate model restricted to function space $H$.

**Theorem 4.6** (Monotonicity under Nested Surrogate Spaces). *Let $H_1 \subset H_2 \subset L^2(X_i)$. Then, for any feature $i$, $S_i^{H_1} \leq S_i^{H_2} \leq S_i^0$.*

The proof is provided in Appendix A.3.

**Corollary 4.7** (Linear vs. Piecewise Linear Surrogates). *Let $L$ denote the linear function space and $P$ a piecewise linear function space such that $L \subset P$. Then $S_i^L \leq S_i^P \leq S_i^0$.*

When $\hat{\theta}_j = 1$ for all $j$, the feature importance induced by surrogate space $H$ satisfies $\hat{I}_i(x_0) = \sqrt{S_i^H} / \sum_{j=1}^p \sqrt{S_j^H}$. As the surrogate space $H$ becomes richer, this quantity approaches the true local importance $\sqrt{S_i^0} / \sum_{j=1}^p \sqrt{S_j^0}$.

When the local behavior of a black-box model exhibits nonlinear effects, piecewise linear surrogate models are more effective than linear surrogates in capturing the importance of features and providing more accurate local explanations. Building on this, we incorporate feature-wise shrinkage weights $\hat{\theta}_i$ to obtain a sparse variance-based importance measure.

This section analyzes variance-based local feature importance from an $L^2$-projection perspective under different surrogate function spaces. Together with the consistency result in Appendix B.3, this analysis helps explain why piecewise-linear surrogates can provide sparse and faithful local feature-importance explanations.

## 5. Experiments

### 5.1. Experimental Setup

We evaluate PL-LIME on both synthetic and real-world tabular datasets to assess its faithfulness and stability. Synthetic experiments provide access to ground-truth feature relevance and importance rankings, enabling an evaluation of explanation correctness. Real-world datasets complement this analysis by assessing explanation reliability through fidelity and stability metrics in the absence of ground truth.

**Synthetic Data Generation.** We construct a synthetic regression task with controllable nonlinearity. The full feature space has 20 dimensions, including correlated distractors and independent noise features, among which four are truly relevant: features 1 and 3 have linear effects, while features 2 and 4 exhibit nonlinear effects. The ground-truth feature importance ranking is predefined during data generation. We use a random forest classifier as the black-box model. Increasing the nonlinearity of the data-generating process induces progressively more nonlinear local behavior in the black-box model, enabling an evaluation of explanation methods under varying degrees of local nonlinearity.

**Real-World Datasets and Black-Box Models.** We evaluate on six UCI tabular datasets: Breast Cancer (569, 30), Ionosphere (350, 34), Wine (178, 13), Adult (32,561, 15), Iris (150, 4), and Bike Sharing (731,11), where each pair denotes the number of instances and features. These datasets cover diverse domains and dimensionalities. For each dataset, we consider two primary black-box models: Random Forest and XGBoost. In additional experiments, we also evaluate neural-network black-box models on three representative datasets. During evaluation, the black-box model hyperparameters and random seeds are fixed across all experiments.

For classification datasets, explanations are constructed for the black-box model's class-0 prediction scores. For the Bike Sharing regression dataset, explanations are constructed for the predicted response. Data are split into training and test sets with an 80/20 ratio, and local explanations are generated for up to 50 randomly selected test instances per dataset; if the test set contains fewer than 50 instances, all test instances are used.

Unless otherwise specified, PL-LIME uses one instance-anchored knot per continuous feature. Appendix D.1 reports a preliminary experiment with additional knots.

**Perturbation Sampling and Weighting.** All compared local explanation methods use the same perturbation generation and weighting scheme. We largely follow LIME's sampling procedure, except that a truncated Gaussian distribution is used to avoid generating perturbations far from the explained instance. Robustness under alternative perturbation sampling strategies is evaluated in Appendix D.3.

Locality is enforced through kernel-based weighting on standardized perturbations, with higher weights assigned to samples closer to the explained instance. We use LIME's default kernel width, $\sqrt{d} \times 0.75$, where $d$ is the number of perturbed features, for both LIME and PL-LIME. A preliminary kernel-width sensitivity check is reported in Appendix D.2.

**Feature Selection** While LIME supports multiple feature selection strategies, we adopt its LASSO-path option, selecting a feature subset from the LARS-computed LASSO path with support size at most $k$.

For fair comparison, we use the same target number $k$ of retained features across methods. For PL-LIME, we follow the solution path of the group nonnegative garrote, as in (Yuan & Lin, 2007; Yuan, 2007), and select the first solution along the path whose active set size equals $k$. When $k$ is not specified in practice, the regularization parameter $\lambda$ can be selected automatically using AIC, BIC, or cross-validation. Appendix D.4 compares these criteria.

**Evaluation Metric** For synthetic experiments, we evaluate explanation quality using two metrics. Let $F_{\text{true}}$ denote the set of ground-truth relevant features and $F$ the set of features selected by the explanation method. The *True Feature Recall* is defined as $\text{TFR} = \frac{|F_{\text{true}} \cap F|}{|F_{\text{true}}|}$. To evaluate ranking accuracy, we measure *Ranking Consistency* by computing the cosine similarity between the ground-truth feature importance ranking vector $\mathbf{r}_{\text{true}}$ and the ranking vector $\mathbf{r}$: $\text{RC} = \frac{\mathbf{r}_{\text{true}} \cdot \mathbf{r}}{\|\mathbf{r}_{\text{true}}\| \|\mathbf{r}\|}$. For real-world datasets, where ground-truth feature relevance is unavailable, we evaluate explanation quality using local fidelity and stability metrics. Local fidelity is measured by a locality-weighted $R_w^2$ between the black-box and surrogate predictions. Stability is evaluated in terms of feature selection stability (Jaccard index) and feature ranking stability (cosine similarity) across repeated explanation runs.

Additional details on data generation, perturbation sampling, weighting scheme and metrics are provided in Appendix C.1.

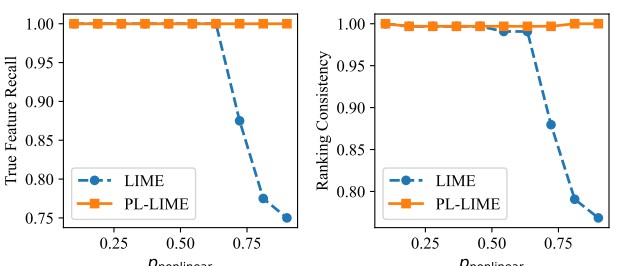

*Figure 1.* Explanation performance on the synthetic dataset under increasing nonlinearity. True feature recall and ranking consistency are reported as functions of the nonlinearity parameter $p_{\text{nonlinear}}$.

## 5.2. Synthetic Experiments: Faithfulness under Increasing Nonlinearity

We use the feature-wise mean vector of each generated dataset as a representative instance for local explanation.

As the degree of nonlinearity increases, the random forest black-box model consistently achieves an AUC of 1, indicating that it fully captures the nonlinear patterns in the data. With increasing nonlinearity, linear surrogate explanations suffer substantial degradation in feature identification and ranking consistency. In contrast, PL-LIME remains robust and consistently recovers both true features and their relative importance, as shown in Figure 1.

We next examine how sparsification interacts with nonlinear local surrogates.

## 5.3. Effect of Sparsification Strategy

**Synthetic Ablation: Coupled Sparsity Distorts Local Structure.** To assess the necessity of decoupled sparsification, We then replace our two-stage strategy with a coupled formulation by adding a group Lasso penalty directly to the first-stage local effect fitting objective. The corresponding objective function is given in Appendix C.2. As the sparsity penalty $\lambda$ increases from $0.01$ to $0.15$, As shown in Figure 6 in the appendix, feature effects that originally exhibit quadratic nonlinearity become progressively flatter. This shows that imposing sparsity during local effect estimation couples feature selection with effect estimation and distorts genuine nonlinear behavior around the explained instance. By contrast, applying sparsity after local effects are estimated preserves local structure while enabling feature selection, which justifies the design in Desideratum D1.

**Real-World Ablation: Disentangling Nonlinear Local Estimation and Sparsification**

We conduct an ablation study to disentangle the contribution of nonlinear local effect estimation from that of decoupled sparsification. Specifically, we construct a nonlinear variant that combines the first-stage local effect estimation of PL-LIME with LIME's original feature selection strategy (Nonlinear-LIME), and compare it against both LIME and PL-LIME. Introducing nonlinear local estimation improves average local fidelity, yielding the ordering PL-LIME $>$ Nonlinear-LIME $>$ LIME. In terms of explanation stability, Nonlinear-LIME exhibits feature selection and feature ranking stability largely comparable to that of LIME, whereas PL-LIME achieves higher stability in both feature selection and feature ranking, as shown in Appendix Figures 7, 8, 9 and 10.

## 5.4. Real-World Experiments: Local Fidelity and Stability

We analyze the results on real-world datasets in terms of local fidelity and explanation stability.

**Local Fidelity.** Figure 2 compares LIME and PL-LIME on six datasets with varying feature dimensionalities in terms of local fidelity ($R_w^2$). Across all datasets and feature counts,

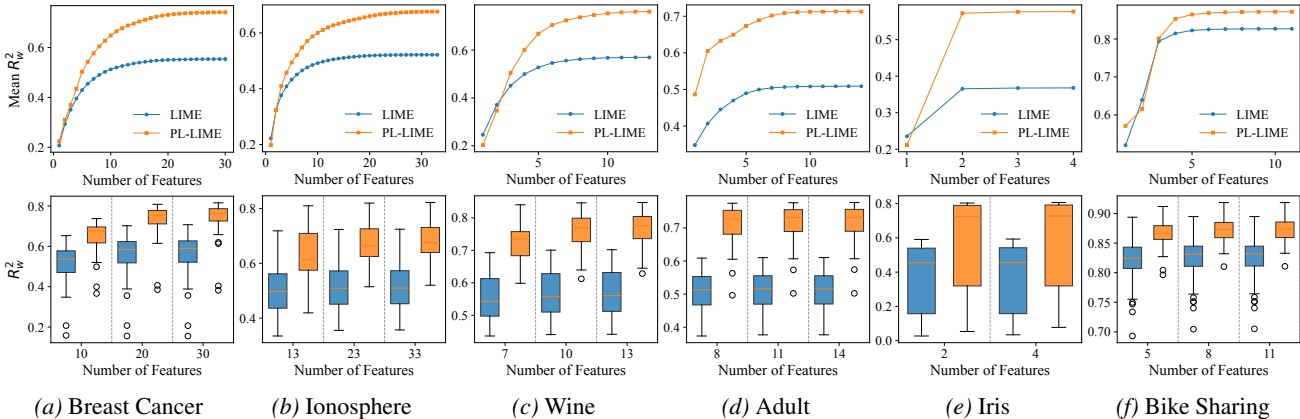

*Figure 2.* Comparison of LIME and PL-LIME across six datasets for explanations of random forest black-box models. Each subfigure corresponds to one dataset and reports the mean local fidelity ($R_w^2$) as a function of the number of features (top) and the distribution of instance-level $R_w^2$ values (bottom) for three different numbers of retained features. The mean local fidelity is computed by averaging $R_w^2$ over a randomly selected set of explained instances, where for each instance $R_w^2$ is obtained by averaging over 50 repeated explanation runs

*Table 1.* Explanation stability (mean values over the selected set of explained instances) under different sparsity levels on real-world datasets for explanations of random forest black-box models. FS denotes feature selection stability and Rank denotes feature ranking stability. Low, medium, and high sparsity correspond to retaining approximately 30%, 60%, and 90% of the total features, respectively.

| SPARSITY | METRIC / METHOD | BREAST CANCER | IONOSPHERE | WINE | ADULT | IRIS | BIKE SHARING |
|---|---|---|---|---|---|---|---|
| HIGH | FS (LIME) | 0.896 | 0.863 | 0.925 | 0.948 | 0.914 | 0.974 |
| HIGH | FS (PL-LIME) | 0.924 | 0.903 | 0.981 | 0.920 | 0.923 | 0.989 |
| HIGH | RANK (LIME) | 0.791 | 0.734 | 0.964 | 0.939 | 0.983 | 0.997 |
| HIGH | RANK (PL-LIME) | 0.833 | 0.752 | 0.979 | 0.959 | 0.985 | 0.997 |
| MEDIUM | FS (LIME) | 0.869 | 0.825 | 0.917 | 0.861 | 0.967 | 0.932 |
| MEDIUM | FS (PL-LIME) | 0.908 | 0.866 | 0.948 | 0.902 | 0.987 | 0.922 |
| MEDIUM | RANK (LIME) | 0.458 | 0.378 | 0.853 | 0.721 | 0.953 | 0.896 |
| MEDIUM | RANK (PL-LIME) | 0.553 | 0.418 | 0.905 | 0.795 | 0.978 | 0.954 |
| LOW | FS (LIME) | 0.888 | 0.877 | 0.920 | 0.899 | 0.925 | 0.892 |
| LOW | FS (PL-LIME) | 0.908 | 0.902 | 0.944 | 0.905 | 0.943 | 0.873 |
| LOW | RANK (LIME) | 0.313 | 0.307 | 0.691 | 0.331 | 0.890 | 0.681 |
| LOW | RANK (PL-LIME) | 0.398 | 0.346 | 0.771 | 0.500 | 0.920 | 0.843 |

PL-LIME consistently achieves higher local fidelity than LIME.

**Stability.** From Table 1, both methods exhibit high feature selection stability (FS), with scores exceeding 0.8 across most datasets and sparsity settings. Overall, PL-LIME achieves slightly higher FS on Breast Cancer, Ionosphere, Wine, Adult, and Iris under most sparsity levels, with only minor exceptions on the Adult dataset under high sparsity and on the Bike Sharing dataset under medium and low sparsity.

In contrast, PL-LIME shows a clearer advantage in feature ranking stability (Rank). Except for the Bike Sharing dataset, where performance is comparable at low sparsity, PL-LIME consistently improves Rank across low, medium, and high sparsity settings on all other datasets, with par-

ticularly pronounced gains under low and medium sparsity. For instance, under low sparsity on Adult, Rank increases from 0.331 to 0.500 (+51%), and under medium sparsity on Breast Cancer, Rank improves from 0.458 to 0.553 (+20.7%).

Additional analyses are provided in the appendix. Appendix C.3 analyzes the observed stability differences. Additional results with XGBoost and neural-network black-box models are reported in Figure 11, Table 2, and Appendix C.4. Appendix C.5 further compares PL-LIME with representative nonlinear local surrogate models.

**Qualitative Observations.**

Beyond quantitative gains in fidelity and stability, PL-LIME also provides feature-level local shape functions. As illustrated in Figure 3, these shape functions capture value-

dependent local response structures, including a sharp transition near a decision threshold on Breast Cancer and a non-monotonic temperature effect on Bike Sharing. These local patterns are averaged out by LIME's linear surrogate. In the Bike Sharing example, around the explained instance, which lies in a relatively high-temperature region, the feature-level local shape function captures a non-monotonic temperature effect that is consistent with the trends shown by the PDP and SHAP dependence plots. Additional details are provided in Appendix C.6.

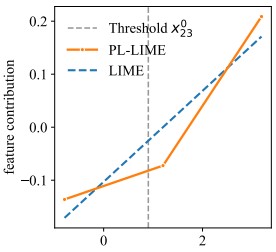 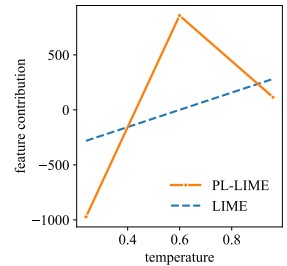

*Figure 3.* Qualitative local explanations produced by PL-LIME and LIME. **Left:** Breast Cancer dataset (instance 204, worst area, Random Forest), where PL-LIME captures a sharp local increase near the decision threshold $x_i^0$ that is missed by LIME's linear approximation. **Right:** Bike Sharing dataset (instance 164, temperature, XGBoost), where PL-LIME reveals a non-monotonic local effect with a trend reversal at high temperatures, while LIME produces a monotonic increasing fit.

## 6. Conclusion and Limitations

In this work, we propose PL-LIME, a sparse piecewise linear local explanation framework for tabular black-box models. PL-LIME anchors the surrogate at the instance being explained and extends linear explanations to two local segments. This design refines the notion of locality by modeling value-dependent feature effects within the local neighborhood. PL-LIME separates local effect estimation from sparsity enforcement through a two-stage design. The decoupled nonnegative shrinkage procedure aligns feature selection with the local surrogate objective and avoids the distortion of local effect shapes caused by coupled sparsification. In addition, the sparse variance-based feature importance measure provides sparse feature rankings. The feature-level local shape functions further offer qualitative descriptions of local response structure. Experiments on synthetic and real-world datasets show improved local fidelity and stability.

PL-LIME has two main limitations. First, it is more suitable for interpretable feature spaces with small or moderate dimensionality, since its computational and memory costs increase with the number of features. Direct application

to raw inputs with very high dimensionality may therefore require feature aggregation, dimensionality reduction, or local feature screening as a preprocessing step. Second, this work uses a single knot anchored at the instance being explained as the default configuration, which provides a practical choice for balancing fidelity, stability, and interpretability. In local regions with stronger heterogeneity or nonlinearity, additional knots may be useful when the added explanation complexity remains acceptable. Developing a principled adaptive strategy for selecting the number and locations of knots according to local model complexity is left for future work.

## Software and Data

The initial implementation and experimental data are available at `https://github.com/goug96/pl-lime`. The repository will be maintained and updated as the project evolves.

## Acknowledgements

This work was supported by the National Natural Science Foundation of China under Grant Nos. 72271171 and 62350062, and Sichuan University's "Double High" Plan for Philosophy and Social Sciences (No. sgjh-202622).

## Impact Statement

This paper contributes to the field of interpretable machine learning by proposing a more faithful and stable local explanation framework for black-box models. The primary impact of this work is methodological, providing improved tools for understanding and analyzing model behavior. Such advances may support more reliable use of machine learning systems, particularly in application domains where transparency and trust are important. We do not foresee any direct negative societal or ethical consequences arising from this work beyond those commonly associated with the broader use of machine learning technologies.

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

# A. Proof

## A.1. Equivalence with Degree-1 Truncated Power Basis Splines

**Equivalence to truncated power basis splines.** We show that the proposed interval-wise parameterization spans the same function class as degree-1 truncated power basis splines. Consider the piecewise linear construction of $u_i(x_i)$ using first-order differences $\{\Delta f_{i,j}\}_{j=1}^m$ at knots $\{k_{i,j}\}_{j=0}^m$ with interval lengths $r_{i,j} = k_{i,j} - k_{i,j-1}$. The induced interval-wise slope function can be written as

$$h_i(x_i) = \sum_{j=1}^m \frac{\Delta f_{i,j}}{r_{i,j}} \mathbb{I}_{[k_{i,j-1},\, k_{i,j})}(x_i).$$

On the other hand, a degree-1 truncated power basis spline admits the representation

$$f_i(x_i) = \beta_{i,0} x_i + \sum_{j=1}^{m-1} \beta_{i,j}(x_i - k_{i,j})_+,$$

where $(x)_+ = \max(x, 0)$. The corresponding slope function is given by

$$h_i^{\mathrm{spline}}(x_i) = \beta_{i,0}\mathbb{I}_{[k_{i,0},\, k_{i,m})}(x_i) + \sum_{j=1}^{m-1} \beta_{i,j}\mathbb{I}_{[k_{i,j},\, k_{i,m})}(x_i).$$

By regrouping terms across successive intervals, we obtain

$$h_i^{\mathrm{spline}}(x_i) = \beta_{i,0}\mathbb{I}_{[k_{i,0},\, k_{i,1})}(x_i) + (\beta_{i,0} + \beta_{i,1})\mathbb{I}_{[k_{i,1},\, k_{i,2})}(x_i) + \cdots + \left(\sum_{j=0}^{m-1}\beta_{i,j}\right)\mathbb{I}_{[k_{i,m-1},\, k_{i,m})}(x_i).$$

Comparing the two representations yields the correspondence

$$\frac{\Delta f_{i,1}}{r_{i,1}} = \beta_{i,0}, \quad \frac{\Delta f_{i,2}}{r_{i,2}} = \beta_{i,0} + \beta_{i,1}, \quad \ldots, \quad \frac{\Delta f_{i,m}}{r_{i,m}} = \sum_{j=0}^{m-1}\beta_{i,j}.$$

Thus, the two parameterizations span the same function space. The interval-wise formulation directly associates each parameter $\Delta f_{i,j}$ with the contribution change over a specific interval, thereby enhancing interpretability while preserving the functional properties of degree-1 truncated power basis splines.

## A.2. Detailed Propositions for LIME under Piecewise-Linear Local Effects and The Proof

**Proposition A.1** (Population LIME coefficients under piecewise-linear true effects). *Let $x_i \in \mathbb{R}^p$ be the instance to be explained. LIME generates perturbed samples $Z \sim \mathcal{N}(x_i, \tau^2 I_p)$, $\quad X = Z - x_i \sim \mathcal{N}(0, \tau^2 I_p)$, and assigns Gaussian kernel weights $w(X) = \exp\left(-\frac{\|X\|_2^2}{2\sigma^2}\right).$*

*Consider the population weighted least-squares surrogate without intercept (features are centered):*

$$\beta^* = \arg\min_{\beta \in \mathbb{R}^p} \mathbb{E}\left[w(X)\left(f(X) - \beta^\top X\right)^2\right].$$

*Assume that the true function is additive, $f(X) = \sum_{j=1}^p f_j(X_j)$.*

*Suppose that for a given feature $r$, the true marginal function $f_r(\cdot)$ is continuous and piecewise linear with knots $k_{r,0} < k_{r,1} < \cdots < k_{r,m}$, allowing $k_{r,0} = -\infty$ and $k_{r,m} = +\infty$, and that these intervals partition the support of the kernel-weighted perturbation distribution. Let $s_{r,j}$ denote the constant slope of $f_r$ on $[k_{r,j-1}, k_{r,j})$, i.e., $f_r'(x) = s_{r,j}$ almost everywhere on $[k_{r,j-1}, k_{r,j})$.*

*Then the population LIME coefficient for feature $r$ satisfies*

$$\beta_r^* = \sum_{j=1}^{m} \omega_{r,j}\, s_{r,j},$$

*where the weights $\{\omega_{r,j}\}_{j=1}^m$ form a convex combination,*

$$\omega_{r,j} \geq 0, \qquad \sum_{j=1}^{m} \omega_{r,j} = 1,$$

*and are given by*

$$\omega_{r,j} = \frac{\int_{k_{r,j-1}}^{k_{r,j}} w_r(x)\, p_\tau(x)\, dx}{\sum_{\ell=1}^{m} \int_{k_{r,\ell-1}}^{k_{r,\ell}} w_r(x)\, p_\tau(x)\, dx},$$

*with*

$$w_r(x) = \exp\left(-\frac{x^2}{2\sigma^2}\right), \qquad p_\tau(x) = \frac{1}{\sqrt{2\pi}\tau} \exp\left(-\frac{x^2}{2\tau^2}\right).$$

*In particular, if $f_r$ is locally linear (i.e., all slopes $s_{r,j}$ are equal), then $\beta_r^*$ exactly recovers this common slope. When $f_r$ is nonlinear, $\beta_r^*$ becomes a kernel-weighted average of heterogeneous interval-wise slopes.*

This proposition corresponds to Proposition 4.1 in the main text and provides a more general statement and a rigorous proof.

*Proof.* Let $Z \sim \mathcal{N}(x_i, \tau^2 I_p)$ be the perturbed samples and define the centered perturbation

$$X := Z - x_i \sim \mathcal{N}(0, \tau^2 I_p).$$

Let the Gaussian kernel weight be

$$w(X) = \exp\left(-\frac{\|X\|_2^2}{2\sigma^2}\right).$$

Consider the population weighted least-squares surrogate without intercept (features are centered):

$$\beta^* = \arg\min_{\beta \in \mathbb{R}^p} \mathbb{E}\left[w(X)\big(f(X) - \beta^\top X\big)^2\right].$$

The corresponding normal equation is

$$\Sigma \beta^* = \Gamma,$$

where

$$\Sigma := \mathbb{E}\left[w(X) X X^\top\right], \qquad \Gamma := \mathbb{E}\left[w(X) X f(X)\right].$$

**Step 1: Structure of the weighted Gram matrix.** For $k \neq \ell$,

$$\Sigma_{k\ell} = \mathbb{E}[w(X) X_k X_\ell] = 0,$$

because $w(X)$ is an even function of $X$ while $X_k X_\ell$ is odd in $X_k$ (or in $X_\ell$) when the other coordinate is fixed, and the distribution of $X$ is symmetric.

For diagonal terms, by isotropy of $X \sim \mathcal{N}(0, \tau^2 I_p)$ and the radial form of $w(X)$, all diagonal entries are equal. Define

$$c := \mathbb{E}\left[w(X) X_1^2\right] > 0.$$

Then

$$\Sigma = c I_p, \qquad \Sigma^{-1} = \frac{1}{c} I_p.$$

**Step 2: Reduction to a single coordinate under additivity.** Assume the true function is additive:

$$f(X) = \sum_{j=1}^{p} f_j(X_j).$$

Fix a feature index $r$. Using $\beta^* = \Sigma^{-1}\Gamma$ and $\Sigma = cI_p$, we have

$$\beta_r^* = \frac{\Gamma_r}{c}.$$

Moreover,

$$\Gamma_r = \mathbb{E}[w(X)X_r f(X)] = \sum_{j=1}^{p} \mathbb{E}[w(X)X_r f_j(X_j)].$$

For $j \neq r$, the term $\mathbb{E}[w(X)X_r f_j(X_j)]$ equals 0 by symmetry in $X_r$ (the integrand is odd in $X_r$ while $w(X)f_j(X_j)$ is even in $X_r$). Hence,

$$\Gamma_r = \mathbb{E}[w(X)X_r f_r(X_r)].$$

**Step 3: Cancellation of $(p-1)$-dimensional factors.** Because $X$ has independent coordinates and the weight factorizes as

$$w(X) = \prod_{j=1}^{p} \exp\left(-\frac{X_j^2}{2\sigma^2}\right),$$

both $\Gamma_r$ and $c = \mathbb{E}[w(X)X_r^2]$ decompose into a product of a univariate integral in $X_r$ and an identical $(p-1)$-dimensional factor over $X_{-r}$. These common factors cancel in the ratio $\Gamma_r/c$, yielding the univariate representation

$$\beta_r^* = \frac{\int_{\mathbb{R}} x\, f_r(x)\, w_r(x)\, p_\tau(x)\, dx}{\int_{\mathbb{R}} x^2\, w_r(x)\, p_\tau(x)\, dx},$$

where

$$w_r(x) = \exp\left(-\frac{x^2}{2\sigma^2}\right), \qquad p_\tau(x) = \frac{1}{\sqrt{2\pi\tau}} \exp\left(-\frac{x^2}{2\tau^2}\right).$$

**Step 4: Piecewise linearity implies a weighted average of interval slopes.** Since

$$w_r(x)p_\tau(x) \propto \exp\left(-\frac{x^2}{2}\left(\frac{1}{\sigma^2} + \frac{1}{\tau^2}\right)\right),$$

the normalized density proportional to $w_r(x)p_\tau(x)$ is the density of a centered Gaussian random variable $\widetilde{X} \sim \mathcal{N}(0, \rho^2)$, where

$$\rho^2 = \frac{\sigma^2\tau^2}{\sigma^2 + \tau^2}.$$

Thus the univariate representation can be written as

$$\beta_r^* = \frac{\mathbb{E}[\widetilde{X} f_r(\widetilde{X})]}{\mathbb{E}[\widetilde{X}^2]}.$$

By Stein's lemma,

$$\mathbb{E}[\widetilde{X} f_r(\widetilde{X})] = \rho^2 \mathbb{E}[f_r'(\widetilde{X})].$$

Since $\mathbb{E}[\widetilde{X}^2] = \rho^2$, we have

$$\beta_r^* = \mathbb{E}[f_r'(\widetilde{X})].$$

Because $f_r$ is continuous and piecewise linear, its derivative exists almost everywhere, and $f_r'(x) = s_{r,j}$ on $[k_{r,j-1}, k_{r,j})$. The knots have probability zero under $\widetilde{X}$, so

$$\beta_r^* = \sum_{j=1}^{m} \Pr\left(\widetilde{X} \in [k_{r,j-1}, k_{r,j})\right) s_{r,j}.$$

Equivalently,

$$\beta_r^* = \sum_{j=1}^{m} \omega_{r,j} s_{r,j},$$

where

$$\omega_{r,j} = \frac{\int_{k_{r,j-1}}^{k_{r,j}} w_r(x) p_\tau(x)\, dx}{\sum_{\ell=1}^{m} \int_{k_{r,\ell-1}}^{k_{r,\ell}} w_r(x) p_\tau(x)\, dx}.$$

These weights are nonnegative and sum to one. This completes the proof. $\square$

**Discussion** This result shows that under isotropic Gaussian perturbations and Gaussian kernel weighting, LIME exactly recovers the true linear coefficients for linear components of $f$, while for nonlinear (piecewise linear) components it estimates a weighted average of local slopes around the instance being explained. The above derivation characterizes the population (asymptotic) solution; the finite-sample estimator converges to $\beta^*$ by the law of large numbers under mild regularity conditions.

### A.3. Proof of Theorem 4.6

*Proof.* For surrogate space $H$, $\Pi_H u_i^0$ satisfies

$$\|u_i^0\|_{L^2}^2 = \|\Pi_H u_i^0\|_{L^2}^2 + \|u_i^0 - \Pi_H u_i^0\|_{L^2}^2.$$

Since $H_1 \subseteq H_2$, we have

$$\|u_i^0 - \Pi_{H_2} u_i^0\|_{L^2}^2 \le \|u_i^0 - \Pi_{H_1} u_i^0\|_{L^2}^2.$$

Hence,

$$\|\Pi_{H_1} u_i^0\|_{L^2}^2 \le \|\Pi_{H_2} u_i^0\|_{L^2}^2.$$

Moreover,

$$\|\Pi_{H_2} u_i^0\|_{L^2}^2 \le \|u_i^0\|_{L^2}^2.$$

Therefore,

$$S_i^{H_1} \le S_i^{H_2} \le S_i^0.$$

$\square$

## B. Additional Method Details

### B.1. Additional Analysis of Local Linearity Violations

Given an instance $x_i$, LIME samples a set of perturbed points $Z_i = \{z^{(n)}\}_{n=1}^N$ from its local neighborhood and assigns each sample a proximity weight $w_i(z) = \exp(-\|z - x_i\|_2^2/\sigma^2)$, where $\sigma > 0$ controls the locality of the explanation. Using the weighted samples $\{(z, B(z)) \mid z \in Z_i\}$, LIME fits a sparse linear surrogate model to approximate the behavior of $B$ in the vicinity of $x_i$.

To explicitly examine the validity of the local linearity assumption, we analyze LIME's explanation process from a single-feature perspective. For a given instance $x_i$ and feature $j$, we generate a perturbed sample set $Z_i^{(j)}$ by varying only the $j$-th feature in the local neighborhood of $x_i$ while keeping all other features fixed at their values in $x_i$. Based on the perturbed values of feature $j$ in $Z_i^{(j)}$, we divide the samples into $m$ consecutive, disjoint intervals $I_{ij}(k)$, $k = 1, \ldots, m$, of equal length. Within each interval $I_{ij}(k)$, we fit a weighted one-dimensional linear regression to estimate the local coefficient: $\hat{\beta}_{ij}^{(k)} = \arg\min_\beta \sum_{z \in I_{ij}(k)} w_i(z)\big(B(z) - \beta z_j - \alpha\big)^2$, where $\alpha$ denotes the intercept term, $\beta$ denotes the slope coefficient associated with $z_j$, and $z_j$ denotes the $j$-th feature of $z$. If the local linearity assumption holds for feature $j$ around instance $x_i$, the coefficients $\{\beta_j^i(k)\}_{k=1}^m$ estimated from different intervals should be consistent.

To quantify the degree of local linearity violation for feature $j$, we define the relative interval effect difference as

$$d_{ij} = \frac{\max_k \hat{\beta}_{ij}^{(k)} - \min_k \hat{\beta}_{ij}^{(k)}}{\frac{1}{m} \sum_{k=1}^m |\hat{\beta}_{ij}^{(k)}|}.$$

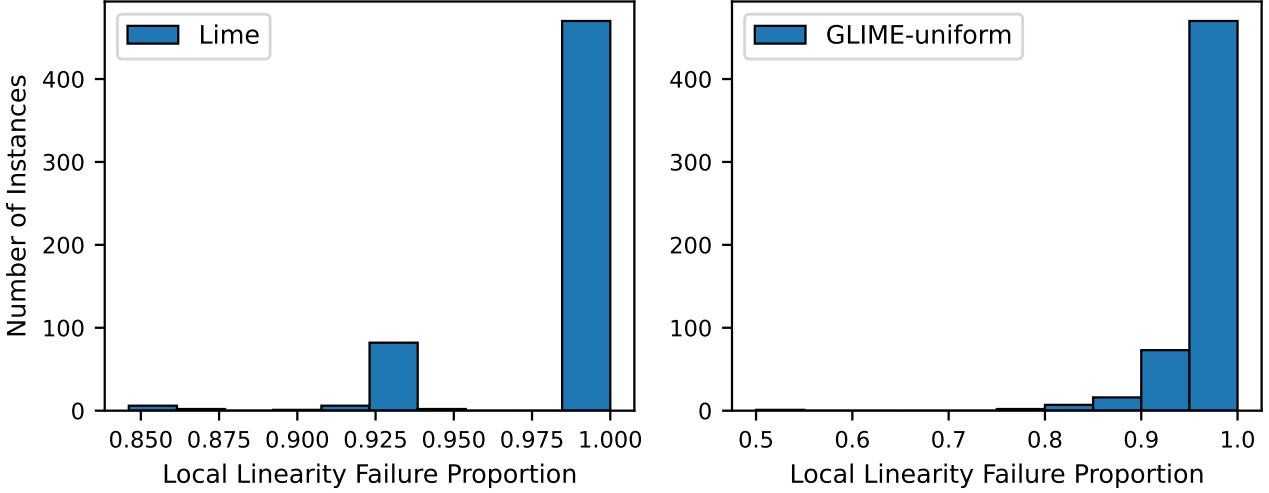

*Figure 4.* Distribution of the local linearity failure proportion across instances on the UCI Breast Cancer dataset using a random forest black-box model. Left: default LIME. Right: GLIME-uniform.

This quantity measures the maximum relative variation of the locally estimated effects across intervals, with larger values indicating stronger deviations from local linearity.

**Definition B.1** (Local Linearity Failure Proportion). For instance $x_i$, let $F$ denote the total number of features. We first identify features with non-negligible local effects by defining a validity mask $\mathcal{M}_{ij} = \mathbb{I}\left(\frac{1}{m}\sum_{k=1}^{m}|\hat{\beta}_{ij}^{(k)}| \geq \epsilon\right)$, where $\epsilon > 0$ is a small threshold. The local linearity failure proportion is then defined as

$$\text{LF}_i = \frac{\sum_{j=1}^{F}\mathcal{M}_{ij}\,\mathbb{I}(d_{ij} > \tau)}{\sum_{j=1}^{F}\mathcal{M}_{ij}},$$

where $\tau > 0$ controls the sensitivity to relative effect variation.

$\text{LF}_i$ represents the proportion of features with meaningful local effects for which the local linearity assumption is significantly violated. This formulation avoids numerical instability when the average local effect magnitude is close to zero.

We conduct experiments on the UCI Breast Cancer dataset. A random forest is used as the black-box model. For continuous features, we consider two perturbed sample generation strategies: the original LIME scheme and an improved variant, GLIME-uniform, which uses a uniform distribution to generate perturbation samples. We set $\epsilon = 10^{-8}$, and $\tau = 1$, indicating that the variation of the locally estimated effects across intervals exceeds their average magnitude. With the number of intervals set to $m = 2$, the feature space is partitioned into two sub-intervals centered at $x_i$, one on each side of $x_i$. then the proposed local linearity failure metric for each instance $x_i$ are computed. The proportions of features whose local effect magnitude $\frac{1}{m}\sum_{k=1}^{m}|\hat{\beta}_{ij}^{(k)}|$ exceeds $\epsilon$ are 0.450 and 0.323 for the two perturbed sample generation strategies, respectively. This indicates that, for this dataset, only a small fraction of features plays an active role in the local neighborhood of the random forest black-box model. We also discuss the failure of LIME when explaining the XGBoost black-box model.

**B.2. Construction of the Design Matrix A**

Let $p_1$ and $p_2$ denote the numbers of continuous and categorical features, respectively. For a given input $\mathbf{x}$, the surrogate prediction can be written as

$$\hat{y} = \sum_{i=1}^{p_1}\theta_i\,\boldsymbol{\alpha}_i(x_i)^{\top}\boldsymbol{\Delta f}_i \;+\; \sum_{i=p_1+1}^{p_1+p_2}\theta_i\,\mathbb{I}\{x_i = x_i^*\}\,\Delta f_i, \tag{3}$$

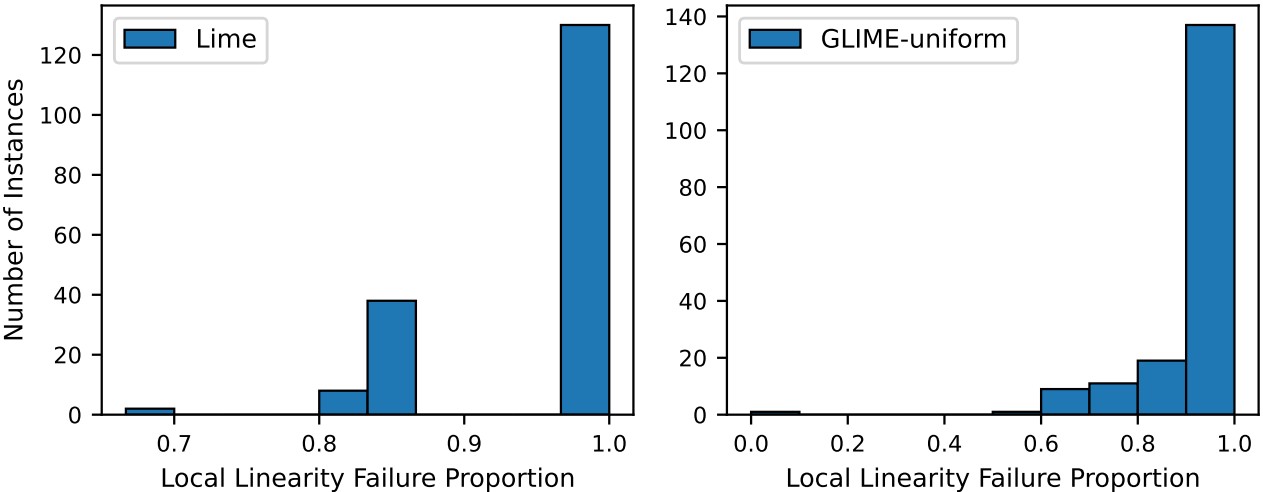

*Figure 5.* Distribution of the local linearity failure proportion across instances on the Wine dataset using an XGBoost model. Left: LIME. Right: GLIME-uniform.

where

$$\boldsymbol{\alpha}_i(x_i) = \big(\alpha_{i1}(x_i), \ldots, \alpha_{im}(x_i)\big)^\top, \qquad \boldsymbol{\Delta f}_i = (\Delta f_{i1}, \ldots, \Delta f_{im})^\top.$$

For continuous features, the basis function $\alpha_{ij}(x_i)$ is defined as

$$\alpha_{ij}(x_i) = \mathbb{I}\{x_i \geq k_{i,j}\} + \frac{x_i - k_{i,j-1}}{r_{i,j}}\mathbb{I}\{k_{i,j-1} \leq x_i < k_{i,j}\},$$

where $r_{i,j} = k_{i,j} - k_{i,j-1}$.

### B.3. Consistency of Sparse Local Explanations

**Corollary B.2** (Consistency of Sparse Local Explanations). *Assume that the initial first-stage local effect estimates $\tilde{u}_1(x_1), \ldots, \tilde{u}_p(x_p)$ satisfy $\ell_2$ consistency, i.e., $\mathbb{E}\big[(u_j(x_j) - \tilde{u}_j(x_j))^2\big] = O_p(\delta_n^2)$, for some $\delta_n \to 0$, where $u_j(x_j)$ denotes the true local effect of feature $j$. If the regularization parameter $\lambda$ tends to zero in a fashion such that $\delta_n = o(\lambda)$, then (i) Feature selection consistency: $P(\hat{\theta}_j = 0) \to 1$ for any $j$ such that $u_j = 0$, and (ii) Estimation consistency: $\sup_j \mathbb{E}\big[(u_j(x_j) - \hat{\theta}_j\tilde{u}_j(x_j))^2\big] = O_p(\lambda^2)$.*

This corollary demonstrates that under the assumption of $\ell_2$ consistency, the proposed sparse local explanation method achieves consistency in both feature selection and estimation. Our second-stage nonnegative shrinkage scheme is closely related to the group nonnegative garrote (Yuan & Lin, 2006; Yuan, 2007), This corollary directly follows from Theorem 2 in (Yuan, 2007).

## C. Additional Experimental Details

### C.1. Experimental Setup Details

**Nonlinear Synthetic Data Generation.** We construct a synthetic dataset with controllable nonlinearity to evaluate the ability of explanation methods to recover ground-truth feature effects. Let $x \in \mathbb{R}^{20}$ denote the input feature vector. Among all features, the first four are ground-truth relevant features, while the remaining features act as distractors. A subset of the distractor features is constructed to be correlated with the true features, and the rest are independent noise variables. In all experiments, the correlation strength between true features and correlated distractors is controlled by the parameter $\alpha_{\text{corr}}$, which is fixed to $\alpha_{\text{corr}} = 0.5$ unless otherwise stated.

To control the degree of nonlinearity, we define a nonlinear transformation function

$$\phi_{\text{nonlinear}}(x) = (1 - p_{\text{nonlinear}})x + p_{\text{nonlinear}}(2x^2 - 1),$$

where $p_{\text{nonlinear}} \in [0, 1]$ governs the strength of nonlinearity. The response variable is generated as

$$y = 2x_1 + \phi_{\text{nonlinear}}(x_2) + 1.5x_3 + 2.5\,\phi_{\text{nonlinear}}(x_4).$$

All features are supported on $[-1, 1]$. Although the response is generated from a regression function, we transform it into a binary classification task when training black-box models, in order to evaluate explanation methods under classification settings. Specifically, the regression outputs are mapped to class probabilities using a logistic transformation.

**Black-Box Model.** All black-box models are trained with a fixed random seed. We use standard off-the-shelf configurations for random forests, XGBoost models, and neural networks, with small neural-network architectures to reduce overfitting. The hyperparameters are kept fixed across repeated runs within each dataset.

**Perturbation Sampling and Weighting Scheme.** For the instance being explained $\mathbf{x}_{\text{expl}}$, with value $\mu_i$ for feature $i$, we generate local perturbations by sampling independently from a truncated normal distribution:

$$x_i \sim \mathcal{N}(\mu_i, \tau_i^2) \text{ truncated to } [\mu_i - 2\tau_i, \mu_i + 2\tau_i],$$

where $\tau_i$ is a feature-specific scale (typically the standard deviation in the training data). The bounds $\pm 2\tau_i$ correspond to truncating a standard normal at $\pm 2$, ensuring samples remain within two standard deviations of the instance while preserving locality. This perturbation scheme restricts samples to a local neighborhood around the explained instance and mitigates the influence of distant samples when approximating local behavior.

**Detailed Evaluation Metrics on Real-World Datasets.** **Local Fidelity.** Local fidelity measures how well the explanation model captures the local decision behavior of the black-box model around an explained instance. In this work, local fidelity is quantified using a weighted coefficient of determination ($R^2$), which evaluates the agreement between black-box predictions and surrogate model predictions under locality-aware weighting. Let $y_i$ and $\hat{y}_i$ denote the predictions of the black-box model and the explanation model for the $i$-th perturbed sample, respectively, and let $w_i$ be the corresponding locality weight. The weighted $R^2$ is defined as $R_w^2 = 1 - \frac{\sum_i w_i(y_i - \hat{y}_i)^2}{\sum_i w_i(y_i - \bar{y}_w)^2}$, where $\bar{y}_w = \frac{\sum_i w_i y_i}{\sum_i w_i}$ is the weighted mean of the black-box predictions. This metric evaluates how faithfully the surrogate captures the local decision behavior of the black-box.

**Stability.** Stable explanations are essential for reliable interpretability. We evaluate stability from two complementary aspects: feature selection stability and feature ranking stability. **Feature Selection Stability.** Feature selection stability measures the consistency of selected feature subsets across repeated explanation experiments. We adopt the Jaccard index to quantify the similarity between feature sets. Let $M$ denote the number of repeated experiments and let $F_{(i)}$ be the feature set selected in the $i$-th experiment. The stability is defined as $\text{Stab}_u = \frac{2}{M(M-1)} \sum_{1 \leq i < j \leq M} \frac{|F_{(i)} \cap F_{(j)}|}{|F_{(i)} \cup F_{(j)}|}$. **Feature Ranking Stability.** Feature ranking stability evaluates the consistency of feature importance rankings across repeated experiments. Let $r_{(i)} \in \mathbb{R}^p$ denote the feature ranking vector obtained in the $i$-th experiment, where $p$ is the total number of features. We employ cosine similarity to measure the similarity between ranking vectors, defined as $\text{Stab}_r = \frac{2}{M(M-1)} \sum_{1 \leq i < j \leq M} \frac{r_{(i)} \cdot r_{(j)}}{\|r_{(i)}\| \|r_{(j)}\|}$.

## C.2. Synthetic Ablation: Effects of Coupled Sparsification

Using the same synthetic dataset, we fix the nonlinearity level to $0.5$ and first estimate nonlinear local feature effects using the first stage of PL-LIME. We replace our two-stage strategy with a coupled formulation by adding a group Lasso penalty to the fitting objective,

$$\min_{\{\Delta f_{i,j}\}} \left\| W^{1/2} \left( \sum_{i=1}^p u_i(x_i) - y \right) \right\|_2^2 + \lambda \sum_{i=1}^p \|\Delta \mathbf{f}_i\|_2, \quad \Delta \mathbf{f}_i = (\Delta f_{i1}, \Delta f_{i2})^\top.$$

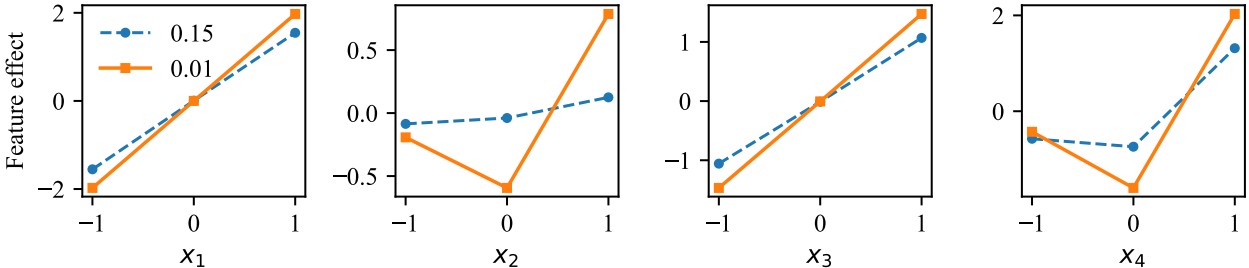

*Figure 6.* Effect of coupled sparsification on local feature effect estimation. As the sparsity penalty $\lambda$ in a group Lasso formulation increases from 0.01 to 0.15, the estimated local feature effect functions around the explained instance become progressively distorted.

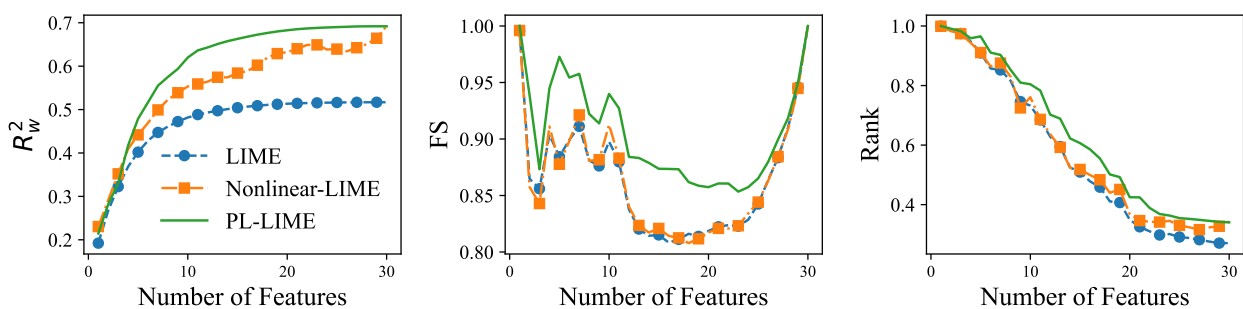

*Figure 7.* Comparison of LIME, Nonlinear LIME, and PL-LIME on the Breast Cancer dataset using a Random Forest black-box model. The figures report local fidelity ($R_w^2$), feature selection stability (FS), and feature ranking stability (Rank) as functions of the number of retained features. Results are averaged over 10 randomly selected instances being explained.

## C.3. Additional Analysis on Stability

For feature selection stability, one possible explanation is that LIME relies on a linear surrogate with feature selection along the LASSO solution path, implicitly assuming locally linear feature effects. When the true local decision function is nonlinear or heterogeneous, systematic residuals may remain, making feature selection more sensitive to perturbation-induced variations. The proposed piecewise linear surrogate partially mitigates this issue by introducing lightweight local nonlinearity, leading to more robust feature selection.

For feature ranking stability, differences persist even under comparable levels of feature selection consistency. Feature importance estimates can be viewed as empirical projections of true local effects onto the surrogate function space, and variability across perturbation runs mainly reflects changes in estimation error directions. Linear surrogates tend to concentrate effects into a small number of correlated directions, increasing sensitivity to perturbations, whereas piecewise linear surrogates distribute effects across multiple local approximations, yielding a smoothing effect that improves ranking stability. This interpretation is consistent with the empirical observations.

## C.4. Results under Neural-Network Black-Box Models

To further examine the applicability of PL-LIME beyond tree-based black-box models, we evaluate it under neural-network black-box models. Since the main experiments focus on tree-based black-box settings, this section considers three representative datasets. Adult represents a tabular classification task, Bike Sharing represents a regression task, and Ionosphere represents a small-scale classification task.

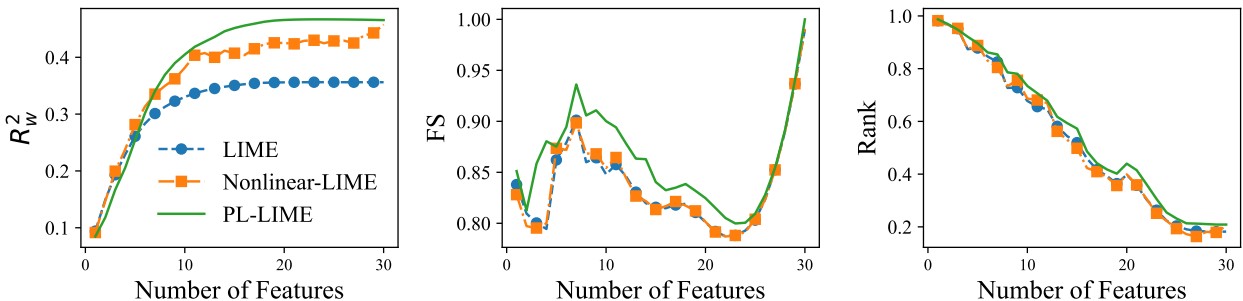

*Figure 8.* Comparison of LIME, Nonlinear LIME, and PL-LIME on the Breast Cancer dataset using an XGBoost black-box model. The figures report local fidelity ($R_w^2$), feature selection stability (FS), and feature ranking stability (Rank) as functions of the number of retained features. Results are averaged over 10 randomly selected instances being explained.

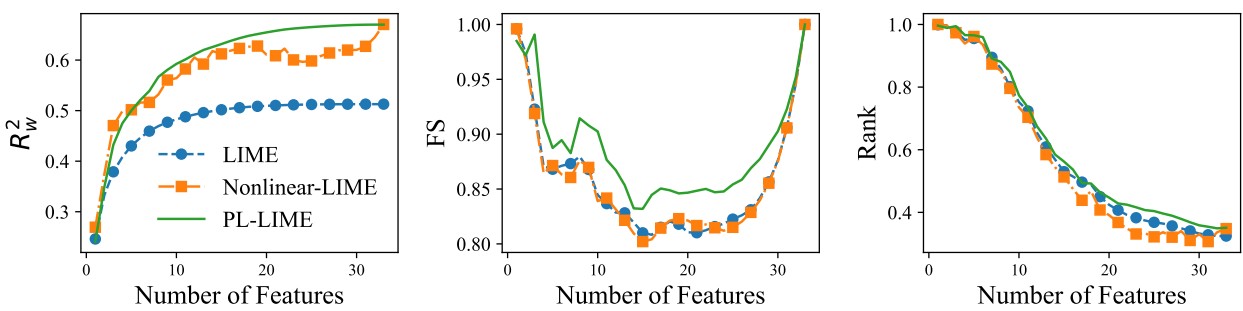

*Figure 9.* Comparison of LIME, Nonlinear LIME, and PL-LIME on the Ionosphere dataset using a Random Forest black-box model. The figures report local fidelity ($R_w^2$), feature selection stability (FS), and feature ranking stability (Rank) as functions of the number of retained features. Results are averaged over 10 randomly selected instances being explained.

Table 3 reports the performance of LIME and PL-LIME under different sparsity levels. Overall, PL-LIME shows favorable fidelity and stability on Adult and Bike Sharing, which suggests that its improvement is not restricted to tree-based black-box models. The results on Ionosphere are more mixed. In terms of fidelity, PL-LIME is slightly better than LIME under medium and low sparsity, but slightly worse under high sparsity. In terms of stability, PL-LIME is comparable to LIME, with slightly lower scores under some sparsity levels. For feature selection, PL-LIME is slightly better than LIME under all three sparsity levels.

To explain the less stable behavior on Ionosphere, we further compare the prediction probability distributions of different black-box models on local perturbation samples. To characterize probability saturation in classification tasks, we compute the distribution of the target-class prediction probabilities $p_i$ and report two statistics, sat_ratio and mid_ratio. Here, sat_ratio denotes the proportion of perturbation samples whose prediction probabilities are close to 0 or 1, while mid_ratio denotes the proportion of perturbation samples whose prediction probabilities fall in the middle range $(0.1, 0.9)$. A higher sat_ratio together with a lower mid_ratio indicates that the local prediction outputs are more saturated, leaving less response variation for the surrogate model to learn.

As shown in Table 4, neural networks produce higher sat_ratio values and lower mid_ratio values than random forests on both Adult and Ionosphere. This pattern is especially clear on Ionosphere. Under the random forest black-box model, sat_ratio is only 0.001 and mid_ratio is 0.969, whereas under the neural-network black-box model, the two values become 0.660 and 0.274, respectively. This indicates that the neural network tends to produce saturated predictions probabilities on

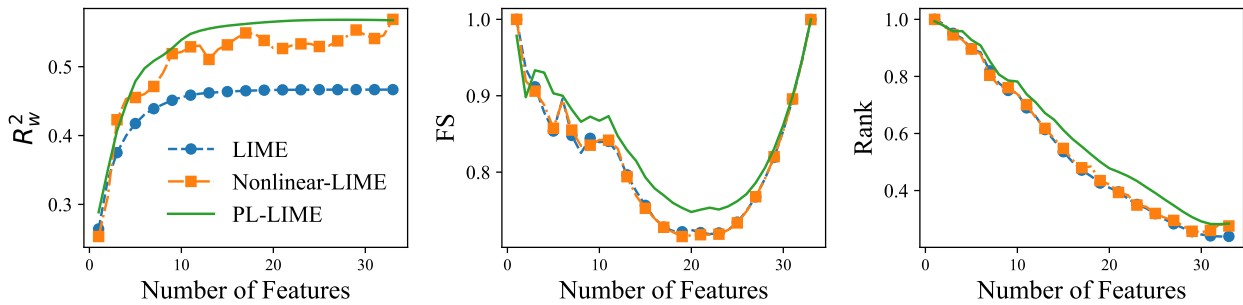

*Figure 10.* Comparison of LIME, Nonlinear LIME, and PL-LIME on the Ionosphere dataset using an XGBoost black-box model. The figures report local fidelity ($R_w^2$), feature selection stability (FS), and feature ranking stability (Rank) as functions of the number of retained features. Results are averaged over 10 randomly selected instances being explained.

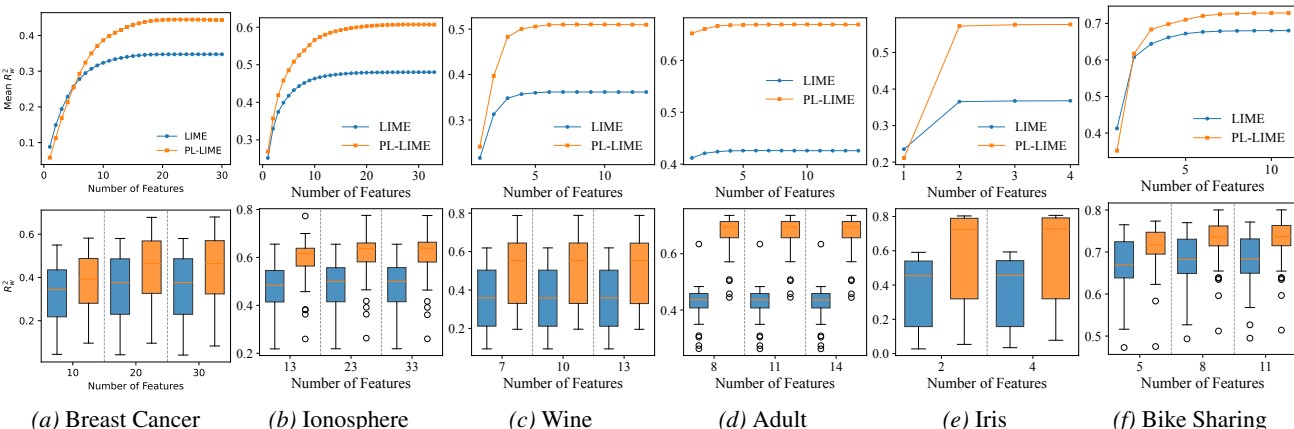

*(a)* Breast Cancer      *(b)* Ionosphere      *(c)* Wine      *(d)* Adult      *(e)* Iris      *(f)* Bike Sharing

*Figure 11.* Comparison of LIME and PL-LIME across six datasets for explanations of XGBoost black-box models. Each subfigure corresponds to one dataset and reports the mean local fidelity ($R_w^2$) as a function of the number of features (top) and the distribution of instance-level $R_w^2$ values (bottom) for three different numbers of retained features.

local perturbation samples from Ionosphere, which reduces the response variation available to the local surrogate model. Therefore, this result does not indicate a failure of PL-LIME itself. Rather, it reflects a common limitation of LIME-style methods that fit local surrogates from prediction confidences when the black-box outputs are highly saturated. Since the advantage of PL-LIME relies on effective response variation in local perturbation samples, extremely saturated local prediction probabilities substantially reduce the learnable variation and weaken the improvement of PL-LIME. In extreme cases, PL-LIME may also be more sensitive to the lack of local response variation.

Overall, the results under neural-network black-box models show that PL-LIME has a certain degree of generalization ability beyond tree-based models. The comparison of local probability distributions also helps explain why the random forest black-box results in the main experiments are more stable.

## C.5. Comparison with Nonlinear Local Surrogate Models

We further compare PL-LIME with two other representative nonlinear local explanation models, BMB-LIME and a decision-tree local surrogate. BMB-LIME is based on Bootstrap Aggregating Multivariate Adaptive Regression Splines Bayesian LIME and uses MARS as its nonlinear local surrogate. BMB-LIME controls model complexity through the number of basis terms and the interaction degree, rather than through explicit feature sparsity. Following (Hung & Lee, 2024), we set the number of basis terms to 20 and the interaction degree to 2, and additionally report results with interaction degree 1. To

*Table 2.* Explanation stability (mean values over the selected set of explained instances) under different sparsity levels on real-world datasets for explanations of XGBoost black-box models. FS denotes feature selection stability and Rank denotes feature ranking stability. High, medium, and low sparsity correspond to retaining approximately 30%, 60%, and 90% of the total features, respectively.

| SPARSITY | METRIC / METHOD | BREAST CANCER | IONOSPHERE | WINE | ADULT | IRIS | BIKE SHARING |
|---|---|---|---|---|---|---|---|
| HIGH | FS (LIME) | 0.862 | 0.842 | 0.946 | 0.784 | 0.999 | 0.949 |
| HIGH | FS (PL-LIME) | 0.885 | 0.877 | 0.957 | 0.781 | 1.000 | 0.944 |
| HIGH | RANK (LIME) | 0.737 | 0.737 | 0.970 | 0.869 | 1.000 | 0.991 |
| HIGH | RANK (PL-LIME) | 0.754 | 0.776 | 0.979 | 0.910 | 1.000 | 0.984 |
| MEDIUM | FS (LIME) | 0.828 | 0.767 | 0.755 | 0.672 | 0.674 | 0.949 |
| MEDIUM | FS (PL-LIME) | 0.852 | 0.803 | 0.792 | 0.656 | 0.719 | 0.944 |
| MEDIUM | RANK (LIME) | 0.425 | 0.462 | 0.785 | 0.485 | 0.669 | 0.991 |
| MEDIUM | RANK (PL-LIME) | 0.455 | 0.515 | 0.822 | 0.584 | 0.735 | 0.984 |
| LOW | FS (LIME) | 0.852 | 0.861 | 0.780 | 0.879 | 0.719 | 0.892 |
| LOW | FS (PL-LIME) | 0.856 | 0.872 | 0.788 | 0.878 | 0.753 | 0.885 |
| LOW | RANK (LIME) | 0.205 | 0.275 | 0.530 | 0.182 | 0.183 | 0.683 |
| LOW | RANK (PL-LIME) | 0.231 | 0.319 | 0.573 | 0.432 | 0.299 | 0.825 |

*Table 3.* Performance comparison between LIME and PL-LIME under neural-network black-box models.

| Dataset | Metric | Method | High sparsity | Medium sparsity | Low sparsity |
|---|---|---|---|---|---|
| Adult | $R_w^2$ | LIME | 0.577 | 0.618 | 0.623 |
| | | PL-LIME | 0.618 | 0.683 | 0.692 |
| | RANK | LIME | 0.933 | 0.751 | 0.508 |
| | | PL-LIME | 0.971 | 0.870 | 0.691 |
| | FS | LIME | 0.950 | 0.873 | 0.900 |
| | | PL-LIME | 0.924 | 0.916 | 0.886 |
| Bike Sharing | $R_w^2$ | LIME | 0.756 | 0.826 | 0.832 |
| | | PL-LIME | 0.815 | 0.869 | 0.887 |
| | RANK | LIME | 0.922 | 0.886 | 0.746 |
| | | PL-LIME | 0.998 | 0.956 | 0.829 |
| | FS | LIME | 0.832 | 0.954 | 0.916 |
| | | PL-LIME | 1.000 | 0.921 | 0.916 |
| Ionosphere | $R_w^2$ | LIME | 0.418 | 0.497 | 0.517 |
| | | PL-LIME | 0.388 | 0.505 | 0.546 |
| | RANK | LIME | 0.648 | 0.387 | 0.347 |
| | | PL-LIME | 0.650 | 0.381 | 0.329 |
| | FS | LIME | 0.856 | 0.856 | 0.921 |
| | | PL-LIME | 0.876 | 0.867 | 0.926 |

ensure a fair comparison, all methods use the same neighborhood perturbation strategy and evaluation procedure, and differ only in the choice of local surrogate model. Since our focus is the behavior of the surrogate model itself rather than the effect of bootstrap aggregation, we do not apply bootstrap aggregation and refer to this variant as MARS-LIME.

In addition to MARS-LIME, we include a decision-tree regressor as a representative tree-based nonlinear local surrogate. For all local surrogate models, the regression target is the black-box confidence score for the target class in classification tasks and the black-box predicted response in regression tasks. Local fidelity is measured by weighted $R^2$. Since many tree-based local explanation methods mainly produce rule-based explanations, which are not directly comparable to the feature-importance vectors considered in this work, we use the built-in feature importance of the decision tree as its explanation vector.

We first report a representative comparison on one classification setting and one regression setting. The goal of this experiment is to illustrate the trade-off among local fidelity, ranking stability, and computational efficiency for different nonlinear local surrogates, rather than to provide an exhaustive benchmark over all nonlinear local explanation methods.

Overall, PL-LIME achieves a more balanced trade-off among fidelity, stability, and efficiency. In both settings, PL-LIME and MARS-LIME achieve high and comparable local fidelity. MARS-LIME shows high stability on Breast Cancer / RF, but much lower stability on Bike Sharing / NN. This difference is related to its effective sparsity. On Breast Cancer / RF,

*Table 4.* Comparison of local prediction probability saturation under different black-box models.

| Dataset | Black-box model | sat_ratio | mid_ratio |
|---------|-----------------|-----------|-----------|
| Adult | Random Forest | 0.093 | 0.795 |
| | Neural Network | 0.459 | 0.413 |
| Ionosphere | Random Forest | 0.001 | 0.969 |
| | Neural Network | 0.660 | 0.274 |

*Table 5.* Representative comparison with nonlinear local surrogate models.

| DATASET / MODEL | METRIC | LIME | PL-LIME | DT-LIME | MARS-LIME |
|-----------------|--------|------|---------|---------|-----------|
| BREAST CANCER / RF | FIDELITY | 0.560 | 0.736 | 0.629 | 0.760 |
| BREAST CANCER / RF | RANK STAB. | 0.298 | 0.388 | 0.321 | 0.790 |
| BREAST CANCER / RF | TIME (MIN) | 6.60 | 10.3 | 10.6 | 76.9 |
| BIKE SHARING / NN | FIDELITY | 0.845 | 0.892 | 0.821 | 0.914 |
| BIKE SHARING / NN | RANK STAB. | 0.748 | 0.841 | 0.924 | 0.381 |
| BIKE SHARING / NN | TIME (MIN) | 0.600 | 1.60 | 1.10 | 17.3 |

MARS-LIME retains only about 9 out of 30 features on average, which makes it substantially sparser than the other methods, so its high stability is partly due to stronger feature selection. On Bike Sharing / NN, MARS-LIME retains about 9 out of 11 features on average, which corresponds to much weaker sparsity and does not lead to a similar stability gain. At the same time, MARS-LIME incurs substantially higher computational cost.

To further separate the effect of surrogate form from the effect of sparsity, we retain the matched-sparsity comparison. In this controlled setting, MARS-LIME and the extended PL-LIME are compared under the same number of retained features, namely 10 features, which corresponds to the typical number of active features selected by MARS-LIME. We evaluate local fidelity, feature selection stability, and computational cost. Table 6 shows that, under the same sparsity constraint, PL-LIME still achieves a more favorable overall trade-off than MARS-based surrogates.

Beyond the numerical results, PL-LIME also has an advantage in the transparency of its explanation form. PL-LIME uses an explicit two-stage procedure. It first estimates feature-level local effects with an instance-anchored piecewise linear model, and then applies feature-level sparsification to these effects through nonnegative scaling factors. As a result, the explanation produced by PL-LIME directly corresponds to feature-level local effect functions and feature importance scores, which makes it easier to analyze how local effects change with feature values.

By contrast, MARS-LIME and the decision-tree surrogate are less direct in two respects. First, in terms of local model construction, MARS-LIME relies on adaptive basis construction and pruning, while the decision-tree surrogate relies on recursive feature splits to partition the local sample space. These structures can increase fitting flexibility, but they do not directly correspond to the local shape of each feature effect in the way PL-LIME does. Second, in terms of feature sparsity or feature-importance generation, the effective feature set of MARS-LIME is determined indirectly by basis selection and pruning, while decision-tree importance is obtained by aggregating impurity reduction over splits. Compared with the explicit feature-level scaling factors in PL-LIME, these mechanisms are harder to interpret as transparent sparsification of local feature effects.

### C.6. Details on Qualitative Local Explanations

This section provides additional details for the qualitative examples shown in Figure 3.

**Local shape functions.**   We provide qualitative examples to illustrate how PL-LIME captures meaningful local structures of black-box models that are missed by LIME.

On the Breast Cancer dataset, we provide local explanations for a Random Forest black-box model with respect to class 0 (malignant). As shown in Figure 3(left), when feature values approach their corresponding threshold $x_{i0}$ (marked by the gray vertical line, where predictions are almost entirely benign for $x_i < x_{i0}$), the shape functions learned by PL-LIME exhibit a sharp increase in malignancy probability around the explanation point. Importantly, both PL-LIME and LIME indicate a positive contribution of worst area to malignancy; however, PL-LIME captures a rapid local change near the

*Table 6.* Comparison of PL-LIME and MARS-LIME under equal sparsity.

| METHOD | LOCAL FIDELITY ↑ | FEATURE SELECTION STABILITY ↑ | RUNTIME ↓ |
|---|---|---|---|
| PL-LIME (INSTANCE-ANCHORED, 1 KNOT) | 0.696 | 1.000 | 22.136 |
| PL-LIME (EXTENDED, 3 KNOTS) | 0.768 | 1.000 | 29.951 |
| PL-LIME (EXTENDED, 5 KNOTS) | 0.783 | 1.000 | 46.303 |
| MARS-LIME (DEG = 1) | 0.755 | 0.967 | 46.148 |
| MARS-LIME (DEG = 2) | 0.774 | 0.954 | 228.861 |

threshold, whereas LIME's linear surrogate only reflects an average positive effect and fails to model this abrupt transition. Figure 3(left) illustrates this behavior for worst area. Overall, PL-LIME provides a more faithful characterization of the local nonlinear behavior of the black-box model.

On the Bike Sharing dataset, we fit an XGBoost black-box model and select instance 164 for local explanation. The temperature feature of this instance takes a value of 0.60 (median: 0.489), corresponding to a moderately high level. As shown in Figure 3(right), for the PL-LIME local fit, the effect of temperature on the predicted rental count is positive to the left of the explanation point, while it becomes negative to the right. This non-monotonic pattern is consistent with real-world observations: moderate increases in temperature generally encourage bike usage, whereas excessively hot conditions tend to suppress demand. In contrast, LIME yields a purely increasing curve and fails to capture this trend reversal. These results indicate that PL-LIME more faithfully characterizes the true local response structure of the model and provides a more expressive local explanation than LIME.

**Consistency with PDP and SHAP Dependence Trends** As a qualitative sanity check, we compare the local shape function learned by PL-LIME with the broader feature-response trends shown by PDP and SHAP dependence plots. This comparison is limited to trend-level consistency, since PL-LIME explains the local neighborhood around a single instance, whereas PDP and SHAP dependence plots describe more global, distribution-level trends.

For the Bike Sharing example, we use the same temperature feature and explained instance as in Figure 3(right), where the temperature value is 0.604. As shown in Figure 12, both the SHAP dependence plot and the PDP show an increase from low to moderate temperature values, followed by a decrease at higher values. This pattern is qualitatively consistent with the PL-LIME shape function in Figure 3(right), which shows a positive local effect to the left of the explained instance and a negative local effect to the right. This agreement should be interpreted as trend-level correspondence rather than pointwise equivalence. Since the explained instance lies near a turning region of the temperature response, such consistency is not unexpected; for instances in other regions, local explanations may reasonably deviate from the global PDP or SHAP dependence trends.

**Auxiliary attribution comparison with SHAP-based methods.** As a further auxiliary reference for the Bike Sharing case study, we additionally compare PL-LIME with SHAP-based attribution methods. SHAP-based methods follow the feature-attribution paradigm rather than the local-surrogate paradigm, and thus we do not directly evaluate SHAP using the weighted local fidelity metric designed for LIME-style methods. Instead, we report normalized Keep AUC, normalized Drop AUC, rank stability, and runtime. This comparison is conducted only on the Bike Sharing dataset, and is intended to provide a limited reference for the case analysis rather than a comprehensive benchmark against SHAP-based methods.

Table 7 reports the results. For the random forest black-box model, TreeSHAP achieves the highest rank stability and the lowest runtime, which is consistent with its specialized design for tree-based models. PL-LIME substantially improves rank stability over LIME while maintaining similar nKA and nDA. For the neural-network black-box model, PL-LIME achieves the highest nKA and nDA and runs much faster than KernelSHAP, although KernelSHAP attains the highest rank stability. Overall, this auxiliary comparison shows that SHAP-based methods are strong attribution baselines, while PL-LIME provides more stable feature rankings than LIME and achieves competitive nKA/nDA with lower runtime than KernelSHAP in the neural-network setting on this dataset.

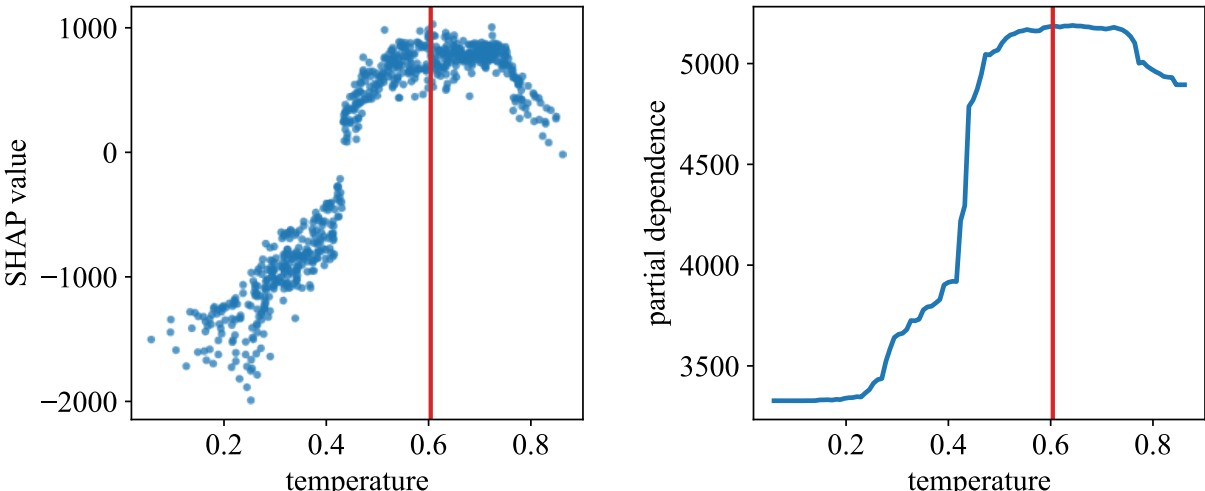

*Figure 12.* Qualitative trend checks for the temperature feature on the Bike Sharing dataset. The left panel shows the SHAP dependence plot, where the contribution of temperature increases from low to moderate values and declines for larger values. The right panel shows the partial dependence plot, where the average predicted rental count shows a similar increase from low to moderate temperatures and weakens at higher temperatures. The red vertical line marks the explained instance used in Figure 3.

*Table 7.* Auxiliary attribution-based check on the Bike Sharing dataset.

| Model | Method | nKA ↑ | nDA ↑ | RS ↑ | RT (min) ↓ |
|-------|--------|-------|-------|------|------------|
| RF | LIME | 0.886 | 0.908 | 0.562 | 2.1 |
| | PL-LIME | 0.866 | 0.907 | 0.779 | 3.5 |
| | TreeSHAP | 0.862 | 0.909 | 1.000 | 1.1 |
| NN | LIME | 0.863 | 0.878 | 0.746 | 0.4 |
| | PL-LIME | 0.868 | 0.889 | 0.828 | 1.1 |
| | KernelSHAP | 0.855 | 0.869 | 1.000 | 7.2 |

## D. Sensitivity and Robustness Analysis

### D.1. Sensitivity to the Number of Knots

We conduct a preliminary sensitivity analysis of the number of knots in PL-LIME using the Breast Cancer dataset with a random forest black-box model. To isolate the effect of knot number, we fix the explanation budget to 20 retained features for all methods. We randomly select 20 instances to be explained and report the average results over these instances. In the main experiments, PL-LIME uses one instance-anchored knot by default, which forms two local linear segments for each continuous feature. This setting introduces only minimal flexibility beyond a linear surrogate while keeping the explanation form simple.

Table 8 reports the results. Under the default one-knot setting, PL-LIME already improves both local fidelity and stability over standard LIME. Under this dataset and black-box setting, further increasing the number of knots improves both local fidelity and stability, but the marginal gains become smaller. At the same time, more knots create more local segments, which can increase the cognitive burden of interpreting the resulting feature effect functions. This suggests that the number of knots can be viewed as a trade-off parameter among fidelity, stability, and interpretability. In practice, a small number of knots often provides a reasonable default. Additional knots can be introduced when the local decision behavior exhibits stronger heterogeneity or nonlinearity and the added explanation complexity remains acceptable.

### D.2. Sensitivity to Kernel Width

We examine the sensitivity of LIME and PL-LIME to the kernel width on the Wine dataset using an XGBoost black-box model. The kernel width is set to $\sqrt{d} \times$ kw, where $d$ is the number of perturbed features and kw is a scaling factor. The

*Table 8.* Effect of the number of knots on the Breast Cancer dataset with a random forest black-box model. The explanation budget is fixed to 20 retained features, and results are averaged over 20 randomly selected explained instances. PL-LIME with one knot corresponds to the default setting used in the main experiments.

| Method | Fidelity ↑ | Rank stability ↑ | Selection stability ↑ |
|---|---|---|---|
| LIME | 0.594 | 0.428 | 0.882 |
| PL-LIME (1 knot, default) | 0.742 | 0.493 | 0.914 |
| PL-LIME (2 knots) | 0.781 | 0.535 | 0.922 |
| PL-LIME (3 knots) | 0.807 | 0.539 | 0.925 |

*Table 9.* Sensitivity to kernel width on the Wine dataset with an XGBoost black-box model. The explanation budget is fixed to 10 retained features, and results are averaged over 20 randomly selected explained instances.

| Metric | Method | 0.25 | 0.5 | 0.75 | 1.0 | 1.25 |
|---|---|---|---|---|---|---|
| $R^2_w$ | PL-LIME | 0.496 | 0.553 | 0.564 | 0.567 | 0.567 |
| $R^2_w$ | LIME | 0.329 | 0.386 | 0.397 | 0.402 | 0.403 |
| FS | PL-LIME | 0.707 | 0.751 | 0.752 | 0.750 | 0.753 |
| FS | LIME | 0.698 | 0.732 | 0.733 | 0.731 | 0.734 |
| RANK | PL-LIME | 0.502 | 0.630 | 0.655 | 0.657 | 0.654 |
| RANK | LIME | 0.470 | 0.582 | 0.593 | 0.594 | 0.598 |

default setting is kw = 0.75. We vary kw over $\{0.25, 0.5, 0.75, 1.0, 1.25\}$, randomly select 20 instances to be explained, and retain 10 selected features for each explanation. The same explained instances and evaluation protocol are used across all kernel-width settings.

Table 9 reports the results. For this dataset and XGBoost black-box model, PL-LIME consistently achieves higher local fidelity, feature selection stability, and feature ranking stability than LIME across all tested values of kw. The local fidelity of both methods increases as kw grows from 0.25 to the default value 0.75, and the results largely saturate when kw is further increased. These results suggest that PL-LIME does not substantially increase sensitivity to kernel width while preserving its advantage over LIME under this setting.

### D.3. Robustness to Perturbation Sampling Strategies

Figure 13 further examines robustness under different perturbation sampling strategies. We consider five perturbation schemes: ITN, IU, MVG, IN, and AE, which denote independent truncated normal, independent uniform, data-driven multivariate Gaussian, independent normal, and autoencoder-based perturbation following ALIME (Shankaranarayana & Runje, 2019). The figure reports local fidelity, feature ranking stability, and feature selection stability.

Under the five perturbation schemes, PL-LIME almost consistently outperforms LIME in local fidelity and stability. The main exception is feature ranking stability on Breast Cancer under MVG perturbations. This behavior is consistent with the strong near-collinearity in this dataset: some feature pairs reach correlations as high as 0.99, and 21 pairs exceed 0.9. Mechanistically, we observe that feature 23 has the highest probability of being ranked as the most important feature, while other features with correlation above 0.9 to feature 23 also appear as the top-ranked feature, although less frequently. This phenomenon is more pronounced in PL-LIME than in LIME, and may therefore lead to lower ranking stability. In contrast, Wine and Ionosphere exhibit milder dependence, with maximum absolute correlations of 0.85 and 0.82, respectively, and fewer highly correlated feature pairs. Under MVG perturbations, PL-LIME still maintains higher ranking stability than LIME on these two datasets. These results suggest that the observed ranking-stability drop may mainly arise under very strong, near-collinear dependence, rather than under feature correlation in general.

### D.4. Sensitivity to automatic $\lambda$ selection.

We further examine the sensitivity of PL-LIME to the automatic selection rule for the regularization parameter $\lambda$. In this experiment, we fix the first-stage local effect estimation and vary only the rule used to select $\lambda$ in the sparsification step. Table 10 compares AIC, BIC, and cross-validation on Wine and Breast Cancer. The last column reports the average number of selected nonzero features.

Overall, PL-LIME is not highly sensitive to the automatic $\lambda$-selection rule. Fidelity remains nearly unchanged across AIC,

*Table 10.* Sensitivity analysis on automatic $\lambda$ selection.

| DATASET | RULE | FID. ↑ | SEL. STAB. ↑ | RANK STAB. ↑ | AVG. # NONZERO ↓ |
|---|---|---|---|---|---|
| WINE | AIC | 0.774 | 0.955 | 0.774 | 12.11 |
| WINE | BIC | 0.773 | 0.955 | 0.771 | 11.93 |
| WINE | CV | 0.773 | 0.955 | 0.764 | 12.07 |
| BREAST CANCER | AIC | 0.752 | 0.912 | 0.402 | 26.37 |
| BREAST CANCER | BIC | 0.752 | 0.904 | 0.389 | 24.89 |
| BREAST CANCER | CV | 0.753 | 0.909 | 0.395 | 26.30 |

BIC, and cross-validation on both datasets, and both feature selection stability and feature ranking stability vary only slightly. BIC tends to produce slightly sparser explanations, especially on Breast Cancer, while maintaining comparable fidelity and stability.

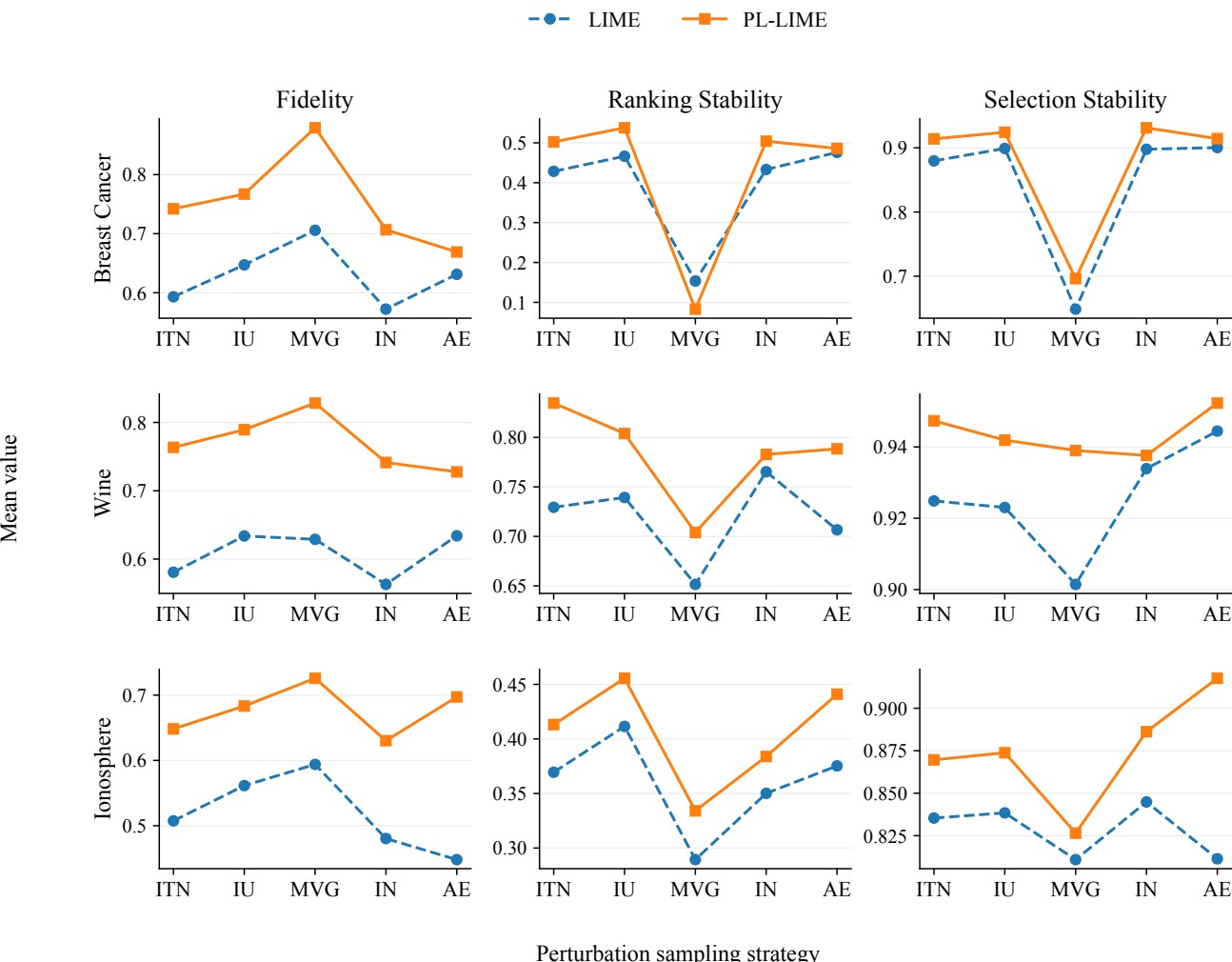

*Figure 13.* Robustness comparison under different perturbation sampling strategies. The rows correspond to Breast Cancer, Wine, and Ionosphere, and the columns correspond to local fidelity, feature ranking stability, and feature selection stability. ITN, IU, MVG, IN, and AE denote independent truncated normal, independent uniform, data-driven multivariate Gaussian, independent normal, and autoencoder-based perturbation, respectively. Breast Cancer and Ionosphere use 20 retained features, while Wine uses 10 retained features.

