# OpenReview forum: "Sparse and Faithful Local Explanations with Piecewise Linear Surrogates"
_ICML.cc/2026/Conference — ICML 2026 regular_

### Official Review · Reviewer_AWeP · 2026-03-08

**Soundness:** 3
**Presentation:** 3
**Significance:** 2
**Originality:** 3
**Overall Recommendation:** 5
**Confidence:** 4

**Summary:**

In this paper, the author has introduce the PL-LIME which is a new variant for LIME algorithm, which performs better explantions than the original LIME algorithm. It enables adding an additoanl knot for each continuous feautres, which makes the local explantion more detailed.

**Compliance With Llm Reviewing Policy:**

Affirmed.

**Final Justification:**

All of my concerns are resolved. I have no further questions towards the paper. I will keep my original score.

**Key Questions For Authors:**

1. How is the Table 3 defined multiple knots?
2. Would the explanation/trend be consistent with partial dependency plot or SHAP? Would it be within the same trend (like up and down within the qualitative examples)?

**Limitations:**

There is no limitation mentioend within the paper, but I do think the selection of knots under a dynamic setup might be important for a piece-wise linear local explantion to be further extended towards a more common use case.

**Strengths And Weaknesses:**

# Strength

**[Clear Idea and motiivation]** The piecewise linear function extends the hypothesis space towards linear functions and make the overall proces more precise in the explantions and thus provides more explanations.

**[Clean theoritical analysis and proofs]** The paper has provided strong theoritical analysis towards the usefulness of the PL LIME and explain why the LIME might fail in some case.

# Weakness

There is no large weakness within the paper. The following are mainly concerns that I have towards the paper

**[Single-knot configuration and Sparse Explanations]** I can understand the single-knots still preserve the simplicity of the explantiosn for the LIME. Even though some explanations on multi-knot has been discussed, I am wondering if the authors has setup some regularzation towards simpler explanations (for instance add a regularization to remove the difference in slope that are relatively small to zero) or when a more complex explanations might need to be made within the sampels.

**[Knot selection in Table 3]** The knot selection in Table 3 for multiple knot seems not to be discussed within the paper.

**[Missing Proof for Thoerem 4.6]** I know this is straightforward, but might worth adding one to make the paper complete.

---

> ### Author Rebuttal · Authors · 2026-03-31
>
> **R4.1 Knot Definition in Table 3**
> In Table 3, multiple knots are defined by placing knots at equally spaced locations within each feature’s range over the generated perturbed samples. Specifically, using \(K\) knots divides the local feature range into \(K+1\) segments, i.e., 1 knot corresponds to 2 piecewise-linear segments, while 3 knots correspond to 4 segments.
>
> **R4.2 Consistency with PDP/SHAP Trends**
> Thank you for the suggestion. Yes. At least for the qualitative examples we examined, the trends are consistent at the qualitative level (e.g., the main up/down pattern and key turning points) with the corresponding partial dependence plots, and are broadly aligned with what one would expect from SHAP dependence plots as well.
>
> Although PL-LIME provides a local, instance-specific shape function, the local response is still influenced by the broader feature-response structure of the black-box model. We therefore use the global feature-shape regularity as a sanity check for the local explanation. In the examples shown, the explained instances happen to lie near important global turning-point regions, so such qualitative agreement is expected.
>
> At the same time, PDP/SHAP dependence plots are more global, whereas PL-LIME is explicitly local. Therefore, exact agreement is not required everywhere, and local explanations for instances in other regions may deviate from the global trend.
>
> **R4.3 Regularization for Simpler Explanations**
> Thank you for the suggestion. At present, we do not impose an additional regularization term specifically for shape smoothing or simplifying slope changes. One main reason is that, in the experiments of this paper, we mainly use the **single-knot configuration**, which yields a **two-segment piecewise-linear explanation** and is already relatively simple. Therefore, we did not further introduce an explicit smoothness constraint. We agree that such regularization becomes more useful when more knots are introduced.
>
> **R4.4 Knot Selection**
> Thank you for the suggestion. Our preliminary results on the Breast Cancer dataset with a Random Forest black-box model (see our response to reviewer 2D3r, R 3.3) show that increasing the number of knots can further improve fidelity (and improve stability), but at the cost of increased explanation complexity. This suggests that the number of knots acts as a **fidelity-interpretability trade-off** parameter: in practice, one may start from the smallest number of knots and increase it only when the gains in fidelity and stability justify the added interpretability burden; once these gains become marginal relative to the increased complexity, the process should stop. Developing a more principled adaptive strategy along this line is an important direction for future work.
>
> **R4.5 Proof Added for Completeness**
> Thank you for the suggestion. We agree that including this proof would make the paper more complete, and we provide it below.
> **Proof.**
> By orthogonality:
> $\|u_i^0\|^2 = \|\Pi_{H} u_i^0\|^2 + \|u_i^0 - \Pi_{H} u_i^0\|^2$.
> Define $S_i^{H} := \|\Pi_{H} u_i^0\|^2$, $S_i^0 := \|u_i^0\|^2$.
>
> $H_1 \subset H_2 \Rightarrow \|u_i^0 - \Pi_{H_2} u_i^0\|^2 \le \|u_i^0 - \Pi_{H_1} u_i^0\|^2$
> $\Rightarrow \|u_i^0\|^2 - \|\Pi_{H_2} u_i^0\|^2 \le \|u_i^0\|^2 - \|\Pi_{H_1} u_i^0\|^2$
> $\Rightarrow \|\Pi_{H_2} u_i^0\|^2 \ge \|\Pi_{H_1} u_i^0\|^2 \Rightarrow S_i^{H_1} \le S_i^{H_2}$.
>
> Also $\|\Pi_{H_2} u_i^0\|^2 \le \|u_i^0\|^2 \Rightarrow S_i^{H_2} \le S_i^0$.
> Hence $S_i^{H_1} \le S_i^{H_2} \le S_i^0$.
>
> **R4.6 Limitations and Future Directions**
> We sincerely thank the reviewer for this valuable suggestion and fully agree that it is an important and promising research direction for future work. Based on this, we summarize several limitations of the current method: (1) Because PL-LIME fits the target-class confidence locally, it may be less informative when this confidence is nearly constant or saturated (close to 0/1) within the neighborhood, leaving little local variation to explain; its greater flexibility may also make it more sensitive to weak local signal than a purely linear surrogate. (2) in practice, explanation design should adaptively balance **fidelity, stability, and interpretability**; and (3) in real-world deployment, both the **number** and **locations** of knots should ideally be selected according to the local complexity of the black-box model. (4) Whether the independent Gaussian assumption for perturbation samples is valid in practice, and how violations of this assumption affect the proposed method, is also an important direction for future work.

---

> > ### Author Rebuttal · Reviewer_AWeP · 2026-03-31
> >
> > Thank the authors for the detailed rebuttals. This fully resolves my concerns. I have no further questions towards the paper, and I will keep the score as it is since it is already an accept. I think it would benefits a lot if the qualitative comparision for PDP and SHAP can be put into the appendix if you have tested that since it would be easier to see if new insights are given from this explanations. That would benefits a lot to the explanablity community.

---

> > > ### Author Response · Authors · 2026-04-01
> > >
> > > Thank you very much for your positive feedback and for recognizing our rebuttal. We are delighted that our response has fully addressed your concerns.
> > >
> > > We also sincerely appreciate your suggestion regarding the qualitative comparison with PDP and SHAP. We fully agree that including such comparisons would help highlight whether additional insights can be obtained from our explanations and would be valuable to the explainability community. As mentioned in our rebuttal, we have conducted qualitative comparisons with PDP and SHAP for the two cases presented in the appendix, and we will include these comparisons in the appendix of the revised manuscript.
> > >
> > > Thank you again for your constructive comments and support.

---

### Official Review · Reviewer_2D3r · 2026-03-10

**Soundness:** 3
**Presentation:** 3
**Significance:** 3
**Originality:** 3
**Overall Recommendation:** 4
**Confidence:** 4

**Summary:**

This paper proposes PL-LIME, a two-stage local explanation framework that addresses two coupled limitations of LIME: the assumption of locally linear feature effects and the misalignment between feature selection and surrogate fitting objectives. The authors investigate the concept of refining locality by modeling each feature's local effect as a piecewise linear function anchored at the explained instance, with a single knot creating two linear segments per feature. Sparsity is enforced through the same surrogate objective rather than through a separate regularized regression. Experiments on synthetic and six real-world UCI datasets demonstrate improvements in local fidelity and explanation stability over LIME.

**Compliance With Llm Reviewing Policy:**

Affirmed.

**Final Justification:**

The paper tackles key limitations of LIME by introducing a novel method that enhances the utility of a well-established interpretability approach. The rebuttal has addressed my primary concerns. The remaining caveat is that the evaluation demonstrates the most substantial gains in application domains where alternative techniques are already readily available.

**Key Questions For Authors:**

1.	How does PL-LIME compare to KernelSHAP in terms of local fidelity and stability on the same datasets?
2.	Have you evaluated PL-LIME on non-tree-based black-box models (e.g., deep neural networks or gradient-boosted models with complex architectures) where TreeSHAP is not available and LIME-style approaches are the primary option?
3.	When are more knots beneficial and are there recommendations for choosing the number of knots?
4.	How sensitive is the method to the choice of perturbation bandwidth, and does the piecewise linear structure mitigate or amplify this sensitivity relative to LIME?

**Limitations:**

yes

**Strengths And Weaknesses:**

**Strengths**

1.	**Well-motivated problem identification.** The paper clearly articulates two coupled structural limitations of LIME: the overly coarse linearity assumption and the misalignment between ℓ₁-based feature selection and ridge regression fitting objectives. The analysis of local linearity violations across datasets (Appendix B.1) provides useful empirical grounding for the proposed approach.

2.	**Theoretically grounded approach.** The piecewise linear surrogate relies on more realistic assumptions about local black-box behavior. For instance, LIME averages over heterogeneous slopes, and PL-LIME captures nonlinear effects better .

3.	**Principled sparsification design.** The non-negative shrinkage approach for obtaining sparse explanations directly addresses the objective misalignment of LIME. By applying multiplicative scaling factors to pre-estimated effects rather than selecting features under a different optimization criterion, the two stages share the same surrogate objective.

4.	**Richer explanations.** Beyond scalar importance scores, PL-LIME provides local effect shape functions that reveal value-dependent feature contributions. The qualitative examples (Figure 11) illustrate meaningful structures — sharp transitions near decision boundaries and non-monotonic effects — that are invisible to LIME's linear surrogates.

5.	**Consistent experimental gains.** Across synthetic and real-world datasets with two black-box models, PL-LIME demonstrates marked advantages over LIME in local fidelity and feature ranking stability, with particularly pronounced improvements under medium and low sparsity settings.


**Weaknesses**

1.	**Missing comparisons to Shapley-based methods.** The paper positions PL-LIME exclusively against LIME and its variants, but does not compare against SHAP or KernelSHAP, which are equally prominent feature attribution methods for local explanations. SHAP provides theoretically grounded importance scores based on marginal contributions and satisfies desirable axiomatic properties (efficiency, symmetry, dummy, additivity). Without this comparison, it is difficult to assess whether PL-LIME's improvements over LIME translate into advantages over the broader landscape of local explanation methods, or whether the identified limitations are specific to LIME's surrogate-fitting paradigm.

2.	**Limited scope of black-box models and datasets raises questions about practical relevance.** The experiments use Random Forest and XGBoost as black-box models on relatively standard UCI datasets. For precisely these model classes, practitioners would typically resort to TreeSHAP, which provides exact Shapley value computation in polynomial time by exploiting the tree structure. LIME (and by extension PL-LIME) is more commonly used as a fallback option when the underlying model is not tree-based — for instance, deep neural networks, large ensembles of heterogeneous models, or proprietary APIs where only prediction access is available. The paper does not demonstrate PL-LIME's performance on these more complex and practically relevant setups. It remains unclear whether the observed fidelity and stability gains persist when explaining, for example, deep networks on higher-dimensional tabular data, or models accessed only through prediction endpoints. Including such scenarios would substantially strengthen the empirical contribution and clarify the practical niche PL-LIME occupies relative to existing tools.

3.	**Number of knots.** The paper fixes the number of knots to one (instance-anchored) throughout the main experiments, which appears to be a modest change compared to LIME.

---

> ### Author Rebuttal · Authors · 2026-03-31
>
> **R3.1 Comparison with KernelSHAP.**
> Thank you for this suggestion. On the Bike Sharing dataset, we have already included a preliminary comparison with SHAP-based methods in another response, see our response to Reviewer a6ZT, Reply R2.3, for the corresponding experimental results, and briefly summarize the key findings here. On the NN model, PL-LIME achieves higher nKA/nDA than KernelSHAP (0.87/0.89 vs. 0.85/0.87) and is substantially more efficient (1.1 min vs. 7.2 min), while KernelSHAP attains higher rank stability (1.00 vs. 0.83).  Overall, these results suggest that PL-LIME offers a better fidelity-efficiency trade-off than KernelSHAP, while KernelSHAP remains stronger in stability. We note that this is only a preliminary comparison on Bike Sharing, and we plan to conduct a larger-scale evaluation with SHAP-based methods in future work.
>
> **R3.2 Evaluation on non-tree black-box models.**
> Yes. We have evaluated PL-LIME on non-tree-based black-box models using MLPs on both **adult** and **MNIST_784**. For adult, we use an MLP with hidden layers (16, 8). For MNIST_784, we first reduce the 784-dimensional pixel input to a 100-dimensional continuous feature space via PCA, and then train an MLP with hidden layers (128, 64, 32). Under different sparsity settings (high, medium, and low) , PL-LIME consistently matches or outperforms LIME in fidelity, rank stability, and feature stability on these non-tree models. This suggests that PL-LIME is applicable beyond tree-based predictors when a meaningful continuous feature space is available.
> | Dataset   | Method  | Fid (H/M/L)    | RankStab (H/M/L) | FeatStab (H/M/L) |
> |-----------|---------|----------------|------------------|------------------|
> | adult     | LIME    | 0.57/0.61/0.61 | 0.91/0.79/0.52   | 0.94/0.87/0.89   |
> | adult     | PL-LIME | 0.60/0.67/0.68 | 0.97/0.91/0.70   | 0.92/0.90/0.90   |
> | mnist_784 | LIME    | 0.37/0.38/0.39 | 0.58/0.32/0.27   | 0.61/0.66/0.84   |
> | mnist_784 | PL-LIME | 0.38/0.41/0.41 | 0.61/0.35/0.30   | 0.62/0.64/0.84   |
>
> **R3.3 Choosing the Number of Knots**
> Thank you for this question. We studied the effect of varying the number of knots on the Breast Cancer dataset with a Random Forest black-box model. Using **1 knot as the default setting**, our preliminary results show that PL-LIME already improves fidelity and stability over standard LIME. Increasing the number of knots further leads to higher fidelity and slightly better stability, although the marginal gains become smaller.
>
> | Method | Fidelity | Rank stability | Selection stability |
> |---|---:|---:|---:|
> | LIME | 0.59 | 0.43 | 0.88 |
> | PL-LIME (1 knot) | 0.74 | 0.49 | 0.91 |
> | PL-LIME (2 knots) | 0.78 | 0.54 | 0.92 |
> | PL-LIME (3 knots) | 0.81 | 0.54 | 0.93 |
>
> Overall, this suggests that the number of knots should be viewed as a **fidelity-interpretability (complexity) trade-off** parameter. In practice, a small number of knots (we use **1 knot as the default**) is often a good starting point, and additional knots can be introduced when higher local fidelity is needed, especially in regions with stronger **local heterogeneity/nonlinearity**, as long as the added complexity remains acceptable. Developing a more principled and adaptive strategy for knot selection is an important direction for future work.
>
> **R3.4 Sensitivity to kernel width.**
> Thank you for this question. We evaluated bandwidth sensitivity on the Wine dataset, where the kernel width is set as $\sqrt{d}\cdot kw$. We varied $kw \in \{0.25, 0.5, 0.75, 1.0, 1.25\}$, randomly selected 20 instances, and fixed the number of non-zero features in the explanation to 10.
>
> As shown below, PL-LIME consistently outperforms LIME across all tested bandwidths. Its variation in fidelity is nearly identical to LIME (Std: 0.027 vs. 0.028), while stability is slightly more sensitive, especially for ranking stability (0.060 vs. 0.049), but only by a small margin. Overall, the piecewise-linear structure does not substantially amplify sensitivity to kernel width.
>
> | Metric | Method | 0.25 | 0.5 | 0.75 | 1.0 | 1.25 | Std. |
> |---|---|---:|---:|---:|---:|---:|---:|
> | Fid. | PL-LIME | 0.50 | 0.55 | 0.56 | 0.57 | 0.57 | 0.027 |
> | Fid. | LIME    | 0.33 | 0.39 | 0.40 | 0.40 | 0.40 | 0.028 |
> | FS   | PL-LIME | 0.71 | 0.75 | 0.75 | 0.75 | 0.75 | 0.018 |
> | FS   | LIME    | 0.70 | 0.73 | 0.73 | 0.73 | 0.73 | 0.014 |
> | RS   | PL-LIME | 0.50 | 0.63 | 0.66 | 0.66 | 0.65 | 0.060 |
> | RS   | LIME    | 0.47 | 0.58 | 0.59 | 0.59 | 0.60 | 0.049 |

---

> > ### Author Rebuttal · Reviewer_2D3r · 2026-04-02
> >
> > Thank you for the clarifications. I will keep my current score, as I believe the overall impact remains modest. The method appears strongest in a domain where strong alternative approaches already exist.

---

> > > ### Author Response · Authors · 2026-04-03
> > >
> > > Thank you for the thoughtful follow-up. We agree that TreeSHAP is a very strong alternative for tree-based models, and we also agree that our current comparison to SHAP-based methods is still limited. We therefore do not intend to position PL-LIME as a replacement for TreeSHAP in settings where exact tree-specific Shapley methods are available.
> > >
> > > Rather, our intended contribution is to the model-agnostic local surrogate explanation setting represented by LIME. In this setting, the goal is not only to assign local importance scores, but also to approximate the local behavior of the black box around the explained instance. From this perspective, we view the paper’s contribution as addressing two structural limitations of LIME within the surrogate framework: (1) the overly restrictive locally linear assumption. Unlike nonlinear surrogate variants that treat the local explanation as a single holistic model, PL-LIME refines locality at the level of individual feature effects through a minimal instance-anchored piecewise-linear parameterization; and (2) the mismatch between feature selection and surrogate fitting objectives, through a sparsification step aligned with the same surrogate objective. In addition, PL-LIME provides instance-anchored piecewise local effect shapes, which can reveal threshold-like or non-monotonic local patterns that are not visible from scalar importance scores alone.
> > >
> > > To clarify that the method is not tied to tree models, we added experiments on non-tree black-box models (MLPs on adult and MNIST_784), where PL-LIME consistently matches or outperforms LIME in fidelity and stability across sparsity settings. While we fully agree that a broader comparison against SHAP-based methods would further strengthen the empirical case, we hope the rebuttal helps clarify that the paper’s main value is as a technically grounded improvement of surrogate-based, model-agnostic local explanations, rather than as a direct replacement for tree-specific Shapley explainers.
> > >
> > > Thank you again for the helpful feedback.

---

### Official Review · Reviewer_a6ZT · 2026-03-11

**Soundness:** 3
**Presentation:** 3
**Significance:** 3
**Originality:** 2
**Overall Recommendation:** 5
**Confidence:** 4

**Summary:**

The paper proposes PL‑LIME, a post‑hoc local explanation framework that replaces LIME’s single linear surrogate with an instance‑anchored piecewise–linear (PWL) surrogate.  Each feature is modeled as a set of linear segments defined by knots that partition the local neighborhood; the simplest instantiation uses one knot per continuous feature, yielding two linear pieces on either side of the explained instance.  A two‑stage sparsification strategy is introduced: first, all PWL effects are estimated by weighted least squares without any sparsity constraints; second, non‑negative shrinkage coefficients (a form of group non‑negative garrote) are fitted to enforce sparsity in a way that is objective‑consistent with the surrogate model.  The authors provide theoretical analysis showing that LIME’s linear coefficient is a kernel‑weighted average of true local slopes, motivating the PWL extension.  Experiments on synthetic data (with controllable nonlinearity) and six UCI tabular datasets with Random Forest / XGBoost black‑boxes demonstrate that PL‑LIME achieves higher local fidelity and improved stability (feature selection & ranking) relative to vanilla LIME, as well as a “non‑linear LIME” baseline that uses the same PWL surrogate but no sparsity.  The paper concludes with a qualitative comparison and discusses limitations.

**Compliance With Llm Reviewing Policy:**

Affirmed.

**Final Justification:**

PL-LIME offers a clean, well-motivated extension of LIME by introducing piecewise-linear surrogates with objective-consistent sparsification. The rebuttal substantially addressed my concerns: the λ-selection ablation (R2.4) confirms robustness to the selection rule, the SHAP comparison (R2.3) is honestly presented, and the correlation analysis (R2.5) clarifies that ranking instability is specific to near-collinear settings. The scalability results (R2.7) and preliminary user study (R2.6) are welcome additions.
Residual weaknesses include a narrow baseline set (no MAPLE, ALIME), modest originality given prior piecewise-linear and sparsity-aware methods, and strong theoretical assumptions. Near-collinearity should be framed as a stated limitation, not only future work.
Overall, the rebuttal changed my assessment from weak reject to weak accept (score 5), conditional on faithful incorporation of the promised revisions.

**Key Questions For Authors:**

1. **Baseline Comparisons**
How does PL‑LIME compare against SHAP (TreeSHAP for tree models) and Anchors on the same datasets?

2. **Hyperparameter λ Selection**
What strategy is used to pick λ in practice (e.g., cross‑validation, stability selection)?

3. **Robustness to Perturbation Distributions**
Theoretical results assume isotropic Gaussian perturbations.  Have you evaluated PL‑LIME under other sampling schemes (e.g., uniform, data‑driven) or correlated features?

4. **Interpretability Evaluation**
Have you conducted any user studies or quantitative metrics (e.g., explanation usefulness, trust) to validate the claim that PL‑LIME’s explanations are more informative?

5. **Scalability Analysis**
What is the runtime and memory footprint of PL‑LIME on datasets with >10,000 features?

**Limitations:**

The authors briefly mention that the method has no direct negative societal impact.  The main limitation is the reliance on strong theoretical assumptions that may not hold in real data, potentially affecting fidelity.  The paper would benefit from a more thorough discussion of these limitations and guidelines for practitioners.

**Strengths And Weaknesses:**

# Soundness
The core ideas are mathematically sound and the proofs in the appendix are rigorous under the stated assumptions. However, key assumptions (additivity of the true function, isotropic Gaussian perturbations, independent features) are strong and not explicitly justified for real‑world tabular data. The two‑stage sparsification scheme is well motivated, but the paper offers limited discussion on how λ is chosen in practice and whether the theoretical consistency results hold under finite‑sample perturbations.

# Presentation
The manuscript is dense with notation and formal definitions, which can be hard to parse on a first read. The narrative flows logically from motivation → theory → method → experiments, but some sections (e.g., the derivation of Proposition 4.1) could benefit from a higher‑level intuition before diving into equations. Figures are clear but more qualitative examples would help readers grasp the practical impact of PWL surrogates.

# Significance
PL‑LIME tackles a well‑known weakness of LIME (inability to capture local heterogeneity) by a minimal extension that is easy to implement. The empirical gains in fidelity and stability are non‑trivial, especially on high‑dimensional datasets. Nonetheless, the contribution is incremental relative to existing non‑linear surrogates (e.g., SHAP‑Tree, MARS‑LIME, BMB‑LIME) and the paper does not fully explore its competitive advantage beyond LIME.

# Originality
The idea of adding a single knot per feature to the surrogate is novel in the context of LIME, and the objective‑consistent sparsification via non‑negative shrinkage is a clever twist. However, piecewise linear surrogates and sparsity‑aware weighting have appeared in related works (e.g., MARS, Anchors), so the novelty is modest. The authors should clarify how PL‑LIME differs from or subsumes these methods.

# Detailed Weaknesses

1. **Strong Theoretical Assumptions** – Additivity, isotropic Gaussian perturbations, and feature independence are rarely satisfied in practice.  The impact of violating these assumptions on the proposed surrogate is not empirically investigated.
2. **Limited Baselines** – The evaluation compares only against vanilla LIME and a “non‑linear LIME” variant.  State‑of‑the‑art methods such as SHAP (TreeSHAP), Anchors, or MARS‑LIME are omitted, making it hard to assess the true competitive edge of PL‑LIME.
3. **Hyperparameter Selection** – The paper mentions a path over λ but does not explain how the final λ is chosen for each instance or dataset.  Without a principled selection rule, the method may be sensitive to λ.
4. **Interpretability Claims** – While the paper claims that PL‑LIME yields more informative local explanations, this is illustrated only qualitatively.  A human‑subject study or quantitative “interpretability” metric would strengthen this claim.
5. **Scalability & High Dimensionality** – Although the method is simple, the paper does not discuss how it scales to thousands of features or very large datasets.  The cost of solving the weighted least squares in the first stage grows with \(p \times m\).
6. **Presentation Clarity** – The heavy use of notation (e.g., \(\Delta f_{i,j}\), \(k_{i,j}\), etc.) can obscure the high‑level ideas.  A schematic diagram of the two‑stage pipeline and a toy example would aid comprehension.

---

> ### Author Rebuttal · Authors · 2026-03-31
>
> **R2.1 Clarity of Presentation.** Thank you for the suggestion. We will improve readability by adding more intuition and schematic diagram of the two-stage pipeline.
>
> **R2.2 Response on baseline selection.**
>  Thank you for the suggestion. Anchors is rule-based and not directly comparable to our feature-based local-surrogate setting. MARS-LIME is already included in the appendix C.4. Additional discussion of nonlinear surrogate variants is provided in our response to **Reviewer fVBD (R1.2)**.
>
> **R2.3 Additional comparison with SHAP / TreeSHAP.**
> Thank you for the suggestion. We agree that SHAP/TreeSHAP is an important reference, though it belongs to a different explanation paradigm (attribution-based rather than local-surrogate). We additionally compare with **TreeSHAP** (for RF) and **KernelSHAP** (for NN) on **Bike Sharing**. Since SHAP is not naturally evaluated by the local-surrogate $R^2$ used for LIME-style explainers, we report **normalized Keep AUC (nKA)**, **normalized Drop AUC (nDA)**, **rank stability (RS)**, and **runtime (RT)**.
>
> | Model | Method | nKA ↑ | nDA ↑ | RS ↑ | RT (min) ↓ |
> |---|---|---:|---:|---:|---:|
> | RF | LIME | **0.89** | 0.91 | 0.56 | 2.1 |
> | RF | PL-LIME | 0.87 | 0.91 | 0.78 | 3.5 |
> | RF | TreeSHAP | 0.86 | **0.91** | **1.00** | **1.1** |
> | NN | LIME | 0.86 | 0.88 | 0.75 | **0.40** |
> | NN | PL-LIME | **0.87** | **0.89** | 0.83 | 1.1 |
> | NN | KernelSHAP | 0.85 | 0.87 | **1.00** | 7.2 |
>
> On Bike Sharing, TreeSHAP is strongest on the RF model and also has the lowest RT, while PL-LIME substantially improves over LIME in RS. On the NN model, PL-LIME achieves the best nKA/nDA and remains much more efficient than KernelSHAP, although KernelSHAP attains the highest RS.
>
> **R2.4 Response on choosing $\lambda$.**
> We compute the **solution path** and select the point with the desired sparsity level. In practice, one can choose the best trade-off among **fidelity**, **stability**, and **interpretability** along the path. Because the complexity of our piecewise-linear surrogate can be quantified by the effective degrees of freedom, **AIC/BIC** are natural options for automatic selection, while **cross-validation** provides a more general data-driven alternative.
>
> **R2.5 Response on robustness to perturbation distributions.**
> We additionally evaluated PL-LIME on the **Breast Cancer** dataset under three perturbation schemes: **independent Gaussian** (the original setting), **uniform**, and a **data-driven multivariate Gaussian** with covariance estimated from the data.
>
> | Sampling | Method | Fidelity ↑ | Selection stability ↑ | Ranking stability ↑ |
> |---|---|---:|---:|---:|
> | Independent Gaussian | LIME | 0.60 | 0.89 | 0.43 |
> |  | PL-LIME | **0.74** | **0.92** | **0.50** |
> | Uniform | LIME | 0.65 | 0.90 | 0.46 |
> |  | PL-LIME | **0.77** | **0.93** | **0.54** |
> | Data-driven MVN | LIME | 0.71 | 0.65 | **0.15** |
> |  | PL-LIME | **0.88** | **0.69** | 0.08 |
>
> Overall, PL-LIME consistently improves fidelity and selection stability across perturbation schemes, while ranking stability is also strong except under the data-driven correlated sampling.
>
> **R2.6 Response on interpretability evaluation.**
> Since explanation usefulness and trust are difficult to assess directly without human evaluation, we instead provide qualitative case studies (Appendix C.5); see our response to **Reviewer fVBD (R1.1)** for discussion of expert feedback. These cases suggest that PL-LIME more faithfully captures the local response structure of the black-box model and yields a richer local explanation than LIME. A dedicated user study is an important direction for future work.
>
> **R2.7Response on scalability.**
> Thank you for the question. We have not yet evaluated PL-LIME beyond \(10{,}000+\) features. As a preliminary check, on **MNIST\_784** with **100/300/500** features, we measured total runtime and peak memory over **5 repeated explanations** of the same randomly selected instance. PL-LIME incurs higher time and memory costs than LIME, and the gap grows with dimensionality:
>
> | #Feat. | LIME time | PL-LIME time | LIME mem. | PL-LIME mem. |
> |--------|----------:|-------------:|----------:|-------------:|
> | 100    | 0.02 min  | 0.10 min     | 110.86 MB | 122.23 MB    |
> | 300    | 0.07 min  | 1.13 min     | 363.20 MB | 788.85 MB    |
> | 500    | 0.11 min  | 4.76 min     | 734.98 MB | 1481.93 MB   |
>
> A more systematic study in the ultra-high-dimensional regime, as well as methods to reduce the computational and memory cost, is left for future work. In fact, we believe local explanation is more meaningful in a lower-dimensional, interpretable feature space than in a very high-dimensional raw input space.
>
> **R2.8 Limitations.** Thank you for raising this point.  Due to space constraints, we refer the reviewer to our response to **Reviewer AWeP, Comment R4.6** for a more detailed discussion of the limitations.

---

> > ### Author Rebuttal · Reviewer_a6ZT · 2026-04-05
> >
> > I thank the authors for the thorough rebuttal and additional experiments. The SHAP/TreeSHAP comparison (R2.3) is informative and honestly presented. I encourage including it prominently in the revision. The commitments on presentation (R2.1) and baseline clarification (R2.2) are satisfactory.
> >
> > ## Remaining concerns:
> >
> > **R2.4** AIC/BIC/CV are mentioned as options but none are empirically validated. A small ablation comparing these strategies across a few datasets would address whether performance is sensitive to the selection rule.
> >
> > **R2.5** Under data-driven MVN sampling, PL-LIME's ranking stability drops below LIME's. Since real-world features are nearly always correlated, this is not an edge case. Could the authors explain why the knot structure amplifies instability under feature dependence?
> >
> > **R2.6** Without a user study or quantitative proxy metric, the claim that PL-LIME explanations are "more informative" rests on case studies. I suggest softening this language in the revision.
> >
> > **R2.7** The ~40× slowdown at 500 features confirms practical limits. Please include this table and a frank discussion of the regime where PL-LIME remains practical.

---

> > > ### Author Response · Authors · 2026-04-07
> > >
> > > **R2.4 Ablation on Automatic λ Selection**
> > >
> > > Thank you for the helpful suggestion. We conducted a small ablation on **Wine Quality** and **Breast Cancer**, comparing the three automatic λ-selection strategies discussed in the paper (**AIC**, **BIC**, and **CV**). We fixed the first-stage basis and varied λ only in the second-stage shrinkage step. The last column reports the average number of selected nonzero features.
> > >
> > > | Dataset | Rule | Fidelity ↑ | Selection Stability ↑ | Ranking Stability ↑ | Avg. # Nonzero |
> > > |---------|------|------------|------------------------|---------------------|------------------|
> > > | Wine | AIC | 0.77 | 0.95 | 0.77 | 12.11 |
> > > |  | BIC | 0.77 | 0.96 | 0.77 | 11.93 |
> > > |  | CV | 0.77 | 0.96 | 0.76 | 12.07 |
> > > | Breast Cancer | AIC | 0.75 | 0.91 | 0.40 | 26.37 |
> > > |  | BIC | 0.75 | 0.90 | 0.39 | 24.89 |
> > > |  | CV | 0.75 | 0.91 | 0.40 | 26.30 |
> > >
> > > Overall, PL-LIME appears **not highly sensitive to the automatic λ-selection rule**: fidelity and stability remain nearly unchanged across **AIC**, **BIC**, and **CV** on both datasets. The main difference is that **BIC** tends to produce slightly sparser explanations with comparable performance.
> > >
> > > **R2.5 Impact of Near‑Collinearity on PL‑LIME’s Ranking Stability**
> > >
> > > Thank you for the insightful question. We agree that feature dependence is common in real-world data. However, our results suggest that the drop in PL-LIME’s ranking stability mainly occurs under **very strong, near-collinear dependence**, rather than under feature correlation in general. In the breast cancer dataset, some feature pairs reach correlations as high as **0.99**, and **21 pairs** exceed **0.9**. In contrast, the wine and adult datasets exhibit milder dependence, with maximum absolute correlations of **0.85** and **0.15**, respectively. Under the same data-driven MVN perturbation, **PL-LIME still achieves higher ranking stability than LIME** on both datasets. We therefore view this issue as mainly arising in near-collinear settings, rather than as a general limitation of PL-LIME under realistic feature dependence.
> > >
> > > | Dataset        | Max. \|corr\| | Ranking stability under data-driven MVN |
> > > |----------------|---------------|-----------------------------------------|
> > > | Breast Cancer  | 0.99          | PL-LIME (0.08) < LIME (0.15)                          |
> > > | Wine           | 0.85          | PL-LIME (0.71) > LIME (0.65)                       |
> > > | Adult          | 0.15          | PL-LIME (0.73) > LIME (0.61)                          |
> > >
> > > Mechanistically, in the breast cancer dataset, we observed that **feature 23 has the highest probability of being ranked as the most important**, while other features with correlation above **0.9** to feature 23 also appear as the top-ranked feature, albeit with lower frequency. This phenomenon is more pronounced in PL-LIME than in LIME.
> > >
> > > This suggests that under near-collinearity, several highly correlated features can explain similar local prediction changes. With its **piecewise linear representation**, PL-LIME allows attribution to shift more flexibly among such features, affecting mainly their **relative ranking** rather than the overall important feature set. In future work, we plan to improve robustness under such dependence via group-level attribution for highly correlated features.
> > >
> > > **R2.6 On the claim of “more informative” explanations**
> > >
> > > Thank you for this helpful suggestion. We will revise the wording to tone down the claim of “more informative.” To further address this concern, we have added a preliminary human-subject study (**17 participants**), which provides initial quantitative evidence that **PL-LIME** outperforms **LIME** in both accuracy and perceived helpfulness, especially for **non-monotonic local effects**. More details are provided in our response to **Reviewer fVBD (Reply Rebuttal Comment R2)**.
> > >
> > > **R2.7 Scalability and Practical Scope**
> > >
> > > Thank you for the suggestion. We will include the runtime and memory results in the revision together with a brief discussion. Based on our current experiments, PL-LIME is most practical for **low- to moderate-dimensional interpretable feature spaces** (e.g.,data with up to a few hundred features), where a single explanation can typically be generated within a few minutes. As dimensionality increases, both runtime and memory usage grow substantially.
> > >
> > > For higher-dimensional settings, PL-LIME is more suitable after preprocessing into a lower-dimensional interpretable representation (e.g., feature aggregation, dimensionality reduction, or local feature screening), similar in spirit to how LIME uses superpixels rather than raw pixels for image explanations. We will clarify in the revision that PL-LIME is currently intended for **moderate-dimensional interpretable spaces**, while systematic evaluation and efficiency improvements for ultra-high-dimensional settings remain future work.

---

### Official Review · Reviewer_fVBD · 2026-03-11

**Soundness:** 3
**Presentation:** 3
**Significance:** 2
**Originality:** 2
**Overall Recommendation:** 3
**Confidence:** 4

**Summary:**

This paper proposes PL-LIME (Piecewise Linear - LIME) , a local post-hoc explanation framework intended to extend LIME, a popular explanation method. While LIME uses a locally linear surrogate function to approximate a non-linear function locally, PL-LIME uses a locally piecewise linear surrogate. A two-stage procedure is devised to compute the PL-LIME surrogates: the method first computes the piecewise linear functions corresponding to each co-ordinate, and then in the second stage, the sparse feature selection is applied. Experiments on synthetic and real-world datasets indicate that PL-LIME achieves higher local fidelity and stability compared to LIME.

**Compliance With Llm Reviewing Policy:**

Affirmed.

**Final Justification:**

Despite the authors' rebuttal, concerns regarding significance remain. The latest response also included a preliminary user study, which contains promising results. I believe the paper can benefit from one additional round of refinement: by providing a more fleshed out user study with clear description of experimental protocols; as well as performing experimentation on neural net models, or at least clearly determining whether the underlying technique is better suited to trees (which are more locally linear), than other model classes.

**Key Questions For Authors:**

Please address questions 1,2,3 in the significance section. In addition, I would be curious to know the authors' stance on:

4. Do you see the present method being applicable to neural network models, and in modalities beyond tabular data?

**Limitations:**

The paper does not discuss any limitations of its approach.

**Strengths And Weaknesses:**

**Soundness**: The paper's arguments are generally sound. The methods are appropriate, and theory seems correct and well-motivated. In particular, I liked the synthetic data setting in the experiments, as it provides a way to compare against ground truth "known" important features. In general, the claims in the paper seem well-supported by experiments and theory.

**Presentation**: The paper is generally written well, and is easy to understand. Related work is also discussed thoroughly.

**Significance**: I believe the paper scores low on significance. While explainability of AI models in general is important, this paper contribution is to improve the mathematical fidelity of an existing popular explanation (LIME), without the broader context of explainability. In general, it is obvious that a more expressive local surrogate strictly improves fidelity, but I believe the key here is not the mathematical properties, but better explanation outcomes, which the paper does not answer, for example:
1. are PL-LIME explanations considered easy to understand, by domain experts? This is important to determine the significance of the approach.
2. do PL-LIME explanations surface novel interactions and effects that other approaches miss? What advantages does the present approach provide other LIME and other non-linear variants of LIME?
3. How does the method perform in comparison to other non-linear LIME surrogates? Line 116-120 in the paper lists other methods that use non-linear surrogates, but it is unclear how the present method performs in comparison to these others, particularly in Figure 1,2 and Table 1. Some limited analysis is performed already in Appendix C.4; but there needs to be more comprehensive comparison of prior works across experiments to demonstrate the strength of the current approach.
4. The scope of the paper is modest: it applies mostly to tabular data, and restricts experimentation to tree-based models such as XGboost and random forests; without showing applicability to other data modalities or to neural network-based models.

**Originality**: The paper scores moderate to low on originality. As mentioned earlier, it is obvious that a more expressive local surrogate strictly improves fidelity, so the key insight is not too novel. I believe the choice of local linearity is well-suited to tree-based models, which could be construed as somewhat novel.

---

> ### Author Rebuttal · Authors · 2026-03-30
>
> **R1.1 Expert interpretability.**
> Thank you for this important question. We believe PL-LIME explanations are easier to understand than standard feature-importance-only explanations because, beyond feature importance, PL-LIME also provides **local shape functions** that show how each feature affects the prediction.
>
> To assess this, we consulted **three experts in management decision-making** and presented these two explanation dimensions. Their feedback suggests that this form is meaningful and useful, as it aligns well with the additive utility view in decision analysis, $y = \sum_i w_i u_i(x_i) $, where $w_i $ reflects feature importance and $u_i(\cdot) $ captures the marginal effect.
>
> For example, in personalized decision-making, one may care not only which factors are important for purchase intention, but also whether the effect of price is monotonic, convex, or exhibits diminishing returns. We believe this intuition also extends to other domains, such as medicine, where an expert may care not only that tumor radius is important, but also whether its effect on risk is linear, thresholded, or diminishing.
>
> A more systematic user study across domains would further strengthen this claim, and we view this as important future work.
>
>  **R1.2 Comparison with nonlinear local surrogate models.**
> We thank the reviewer for this helpful suggestion.
> In addition to MARS-LIME (already included in the appendix C.4), we also evaluated a decision-tree regressor as a local explainer, using the target-class confidence score as the regression target, $R^2$ as fidelity, and built-in tree importance as feature importance. This covers representative nonlinear surrogates from both GAM-based and tree-based families. QLIME and LEMNA showed unsatisfactory performance in our preliminary experiments, while LORE mainly produces rule-based explanations rather than feature-importance vectors, so we do not include them in the main comparison.
>
> PL-LIME achieves the best overall trade-off among fidelity, stability, and efficiency in the main results. While MARS-LIME attains very high stability on Breast Cancer, this mainly comes from much stronger sparsity (10/30 retained features on average), whereas the other methods are nearly non-sparse. Appendix C.4  shows that under the same sparsity level, PL-LIME still yields a better overall trade-off.
>
> | Metric    | LIME | DT   | PL   | MARS | \| | LIME | DT   | PL   | MARS |
> |-----------|------|------|------|------|----|------|------|------|------|
> |           | **Breast Cancer / RF** |  |  |  | \| | **Bike Sharing / NN** |  |  |  |
> | Fidelity  | 0.56 | 0.63 | 0.74 | 0.76 | \| | 0.85 | 0.82 | 0.89 | 0.91 |
> | Stability | 0.30 | 0.32 | 0.39 | 0.77 | \| | 0.75 | 0.92 | 0.83 | 0.37 |
> | Time (min)  | 6.6  | 10.6 | 10.3 | 76.9 | \| | 0.6  | 1.1  | 1.6  | 17.3 |
>
> Beyond the above discussion, we would further clarify that  other advantages of our method are as follows:
>
> **Transparent pipeline.** PL-LIME has a transparent two-stage pipeline: it first fits a piecewise-linear model using first-order differences, then applies feature-level sparsification via nonnegative factors that shrink local effects. This is more transparent than MARS, which relies on basis construction/pruning, or decision trees, which rely on split/pruning rules. We believe such transparency is crucial for local interpretable surrogate models. **Theoretical support.** We provide theoretical justification for the resulting local explanation, supporting its validity.
>
> **R1.3 Interactions.**
> Thank you for the suggestion. For interpretability, PL-LIME does not explicitly model interactions. Interaction-aware extensions are left for future work.
>
> **R1.4 Applicability beyond tabular/tree settings.**
> **Yes.** The method is **model-agnostic** and does not rely on any tree-specific structure, so it is applicable to **neural network models** as well.
> We additionally tested PL-LIME on **MNIST\_784**, an **image** dataset, using a neural predictor. In the same PCA-reduced feature space and under different sparsity settings (high, medium, and low ), PL-LIME consistently matches or outperforms LIME in **fidelity**, **rank stability**, and **feature stability**:
>
> | Dataset   | Method  | Fid (H/M/L)   | RankStab (H/M/L) | FeatStab (H/M/L) |
> |-----------|---------|---------------|------------------|------------------|
> | mnist_784 | LIME    | 0.37/0.38/0.39| 0.58/0.32/0.27   | 0.61/0.66/0.84   |
> | mnist_784 | PL-LIME | 0.38/0.41/0.41| 0.61/0.35/0.30   | 0.62/0.64/0.84   |
>
> This suggests that the method extends beyond **tabular/tree settings**. More generally, it can be applied to other modalities (e.g., images or text) when a suitable **continuous or semantic feature representation** is available.
>
> **R1.5 Limitations.**
> Due to space constraints, we refer the reviewer to our response to **Reviewer AWeP, R4.6** for a more detailed discussion of the limitations.

---

> > ### Author Rebuttal · Reviewer_fVBD · 2026-04-04
> >
> > I thank the authors for their rebuttal, and for the additional experiments. Unfortunately, the main concerns regarding the significance and real-world utility of the approach remain, and hence I am not able to increase my score.

---

> > > ### Author Response · Authors · 2026-04-07
> > >
> > > ### R1. Why PL-LIME Improves Explanations Beyond Fidelity
> > > We agree that flexibility alone is not sufficient; the key question is whether it leads to better explanations. Our point is that **PL-LIME provides several concrete explanation benefits beyond a numerical fidelity gain**.
> > >
> > > (1) PL-LIME provides **more faithful local feature-importance explanations**. In **Section 5.2**, we directly evaluate explanation quality on synthetic experiments with known ground truth using **True Feature Recall** and **consistency with the ground-truth feature ranking**. These results show that under **local nonlinearity**, LIME may produce unfaithful feature importance, whereas **PL-LIME recovers feature importance more faithfully**. This empirical finding is also consistent with the theoretical analysis in **Proposition 4.1** and **Corollary 4.7**.
> > >
> > > (2) PL-LIME captures **finer-grained local black-box behavior**. Unlike nonlinear surrogates that still treat the neighborhood as **a single region**, **PL-LIME is instance-anchored and piecewise linear**, modeling the neighborhood as **finer-grained sub-local regions** centered at the explained instance. It can therefore reveal **thresholds and turning points** that a single linear slope flattens away, as illustrated in **two representative case studies in Appendix C.5**.
> > >
> > > (3) Many existing **nonlinear sparse local explanation methods** still couple sparsity with the fitting objective, which can distort local structure under nonlinearity. By contrast, **Section 5.3** shows that PL-LIME uses a decoupled strategy that separates local effect estimation from sparse feature selection, thereby yielding sparse explanations without sacrificing local structure. **Corollary B.2** further establishes consistency in both feature selection and estimation, and the same explicit two-stage design makes the sparsification mechanism **more transparent than methods that rely on heuristic pruning procedures**.
> > >
> > > To further address the reviewer’s question, we additionally conducted a small questionnaire study.
> > >
> > > ### R2. Preliminary Human Evaluation on Explanation Usefulness
> > > We conducted a small questionnaire study on the bike-sharing rental dataset with **17 participants** to examine whether **PL-LIME can help users understand local feature effects more effectively than LIME**, especially for the **atemp** feature (perceived temperature).
> > >
> > > For the same explanation instance, **LIME** provides the **top-5 important features** and whether the coefficient of **atemp** is positive or negative. **PL-LIME** provides the **top-5 important features** and a **piecewise local shape function for atemp**, consisting of **two linear segments**, one on each side of the explained instance.
> > >
> > > Based on these explanations, we asked participants the following four questions:
> > > 1. how the predicted rental count changes when **atemp** increases;
> > > 2. how the predicted rental count changes when **atemp** decreases;
> > > 3. whether the provided explanation is consistent with their subjective understanding (**1–5 rating, higher is better**);
> > > 4. how helpful the provided explanation is for answering Questions 1 and 2 (**1–5 rating, higher is better**).
> > >
> > > We selected explained instances based on the single-feature local analysis in **Appendix B.1**, where only **atemp** is perturbed around the explained instance while all other features are fixed.
> > > - The first is a **monotonic case** at a **low-temperature** point, where the black-box response increases on both sides of the instance, although the left-side slope is smaller.
> > > - The second is a **turning-point case** at a **medium-to-high temperature** point, where the black-box response increases on the left side but decreases on the right side.
> > >
> > > Among the 17 participants, **9** viewed the **LIME** explanations and **8** viewed the **PL-LIME** explanations. Each participant answered the four questions for both cases. The results are as follows.
> > >
> > > | Method | Accuracy (Monotonic) | Accuracy (Turning-point) | Consistency (Monotonic) | Consistency (Turning-point) | Helpfulness (Monotonic) | Helpfulness (Turning-point) |
> > > |--------|-----------------------|--------------------------|--------------------------|-----------------------------|--------------------------|-----------------------------|
> > > | LIME | 0.89 | 0.70 | 3.33 | 3.22 | 3.11 | 3.22 |
> > > | PL-LIME | 0.94 | 1.00 | 4.25 | 4.00 | 4.38 | 4.25 |
> > >
> > > These results show that, on the monotonic case, the two methods achieve similar objective accuracy, although PL-LIME still receives higher subjective ratings. On the turning-point case, PL-LIME achieves higher accuracy and higher consistency/helpfulness ratings than LIME, suggesting that PL-LIME can better support user understanding when local feature effects are more complex or non-monotonic. This can help bike-rental operators identify temperature ranges where demand responds differently, enabling more informed decisions on pricing, fleet allocation, and staffing.

---

### Decision · Program_Chairs · 2026-04-30

**Decision:**

Accept (regular)

**Comment:**

The paper studies a variant of LIME called PL-LIME, which uses piecewise linear functions to approximate the black box function.

We had a good discussion among ourselves for this paper. While there were concerns about the significance and limited evaluation (on three-based models), the rebuttal partially resolved some of these issues for the reviewers. While I do not advocate for including the user study in the final version (given its small size), I think the neural network model results should be included. There were many good points in the author-reviewer discussion, and I hope the authors include those in the final version. In particular, the authors should clearly state the limitations of the approach as it does not scale, and also include a discussion about the number of knots.